# Ret receptor tyrosine kinase sustains proliferation and tissue maturation in intestinal epithelia

Daniel Perea[1], Jordi Guiu[2], Bruno Hudry[1], Chrysoula Konstantinidou[1], Alexandra Milona[1], Dafni Hadjieconomou[1], Thomas Carroll[1], Nina Hoyer[3], Dipa Natarajan[4], Jukka Kallijärvi[5,†], James A Walker[6], Peter Soba[3], Nikhil Thapar[4], Alan J Burns[4], Kim B Jensen[2,7] & Irene Miguel-Aliaga[1,*]

## Abstract

Expression of the Ret receptor tyrosine kinase is a defining feature of enteric neurons. Its importance is underscored by the effects of its mutation in Hirschsprung disease, leading to absence of gut innervation and severe gastrointestinal symptoms. We report a new and physiologically significant site of Ret expression in the intestine: the intestinal epithelium. Experiments in *Drosophila* indicate that Ret is expressed both by enteric neurons and adult intestinal epithelial progenitors, which require Ret to sustain their proliferation. Mechanistically, Ret is engaged in a positive feedback loop with Wnt/Wingless signalling, modulated by Src and Fak kinases. We find that Ret is also expressed by the developing intestinal epithelium of mice, where its expression is maintained into the adult stage in a subset of enteroendocrine/enterochromaffin cells. Mouse organoid experiments point to an intrinsic role for Ret in promoting epithelial maturation and regulating Wnt signalling. Our findings reveal evolutionary conservation of the positive Ret/Wnt signalling feedback in both developmental and homeostatic contexts. They also suggest an epithelial contribution to *Ret* loss-of-function disorders such as Hirschsprung disease.

**Keywords** *Drosophila*; enteroendocrine; intestine; Ret; stem cell
**Subject Categories** Development & Differentiation; Signal Transduction; Stem Cells
**The EMBO Journal (2017) 36: 3029–3045**

## Introduction

The identification of the Ret receptor over 20 years ago, at the time enabled by its oncogenic potential when ectopically activated, spurred a flurry of research into this highly conserved transmembrane receptor with tyrosine kinase activity (Takahashi *et al*, 1985; Takahashi & Cooper, 1987; Ibanez, 2013). It is now well established that the endogenous activity of Ret is required for the formation and/or maintenance of a variety of cell types including neuronal, kidney, lymphoid, hematopoietic and testis germ cells (Schuchardt *et al*, 1994; Naughton *et al*, 2006; Veiga-Fernandes *et al*, 2007; Ibanez, 2013; Fonseca-Pereira *et al*, 2014). In these cells, Ret signalling is involved in a diverse range of developmental processes including proliferation, migration survival and/or differentiation (Sasselli *et al*, 2012; Ibanez, 2013). The importance of maintaining appropriate levels of Ret signalling—and doing so in the right place and at the right time—is highlighted by the severity of disorders resulting from its dysregulation. First found in a human lymphoma (Takahashi *et al*, 1985), ectopic activation of Ret signalling is now a recognised and significant feature of several other cancers (Ibanez, 2013; Santoro & Carlomagno, 2013; Romei *et al*, 2016). The digestive system is the site of perhaps one of the most striking clinical manifestations resulting from *Ret* dysregulation: Hirschsprung disease (or HSCR). Frequently resulting from *Ret* loss-of-function mutation and affecting ca. 1 in 5,000 humans, HSCR leads to a variety of severe gastrointestinal symptoms such as abnormal peristalsis and bowel obstruction, which have been attributed to a striking absence of enteric innervation in the distal part of the gut (Martucciello *et al*, 2000; Sasselli *et al*, 2012). Here, we report an unexpected and important role for Ret in a different gastrointestinal cell population: the intestinal epithelium.

## Results

### Ret is expressed by the adult somatic progenitors of the *Drosophila* intestine

Expression of Ret has been reported in developing neurons of many animals including *Drosophila* (Pachnis *et al*, 1993; Sugaya *et al*,

---

1  MRC London Institute of Medical Sciences, Imperial College London, London, UK
2  BRIC—Biotech Research and Innovation Centre, University of Copenhagen, Copenhagen N, Denmark
3  Center for Molecular Neurobiology, University Medical Center Hamburg-Eppendorf (UKE), University of Hamburg, Hamburg, Germany
4  Stem Cells and Regenerative Medicine, UCL Institute of Child Health, London, UK
5  Institute of Biotechnology, University of Helsinki, Helsinki, Finland
6  Center for Human Genetic Research, Massachusetts General Hospital and Harvard Medical School, Boston, MA, USA
7  The Danish Stem Cell Center (Danstem), Faculty of Health and Medical Sciences, University of Copenhagen, Copenhagen, Denmark
   *Corresponding author. Tel: +44 208 3833907; E-mail: i.miguel-aliaga@imperial.ac.uk
   †Present address: Folkhälsan Research Center, Helsinki, Finland

1994; Hahn & Bishop, 2001; Read *et al*, 2005; Kallijarvi *et al*, 2012; Hernandez *et al*, 2015; Soba *et al*, 2015). Prompted by our interest in the *Drosophila* intestine and its neurons (Cognigni *et al*, 2011; Lemaitre & Miguel-Aliaga, 2013), we investigated a possible conservation of Ret expression in the enteric nervous system of flies. Immunohistochemistry using a Ret antibody and *Ret-Gal4* reporter confirmed expression in central gut-innervating neurons and enteric ganglia, both during development and in adult flies (Fig 1A–D and data not shown). During the course of these experiments, we unexpectedly detected the Ret reporter in the adult midgut (Fig 1A, F and G): a portion of the *Drosophila* intestine analogous to the mammalian small/large intestine, which harbours a self-renewing epithelium (Lemaitre & Miguel-Aliaga, 2013). Expression of Ret in the adult midgut epithelium was confirmed using the Ret-specific antibody (Fig 1E). Co-staining with cell type-specific markers revealed that Ret was generally absent from differentiated epithelial cells (enterocytes and enteroendocrine cells, Fig 1A and G), but was expressed by adult somatic intestinal stem cells (ISCs) and their postmitotic, undifferentiated progeny: the enteroblasts (EBs; Fig 1A, E and F). Hence, in addition to evolutionary conserved expression in enteric neurons, expression analysis of the neurotrophic factor receptor Ret in the *Drosophila* intestine further reveals a previously unrecognised site of Ret production: adult somatic epithelial progenitors.

## Ret sustains stem cell proliferation in the adult *Drosophila* intestine, both in homeostasis and during regeneration

The presence of Ret in adult intestinal progenitors prompted us to investigate possible effects of interfering with *Ret* function on proliferation. We conducted a series of experiments in virgin females, the stem cells of which proliferate more than those of males (Hudry *et al*, 2016) and have not been affected by gut-remodelling mating hormone(s) (Reiff *et al*, 2015). These experiments involved adult intestinal progenitor-specific downregulation of *Ret*, achieved by expression of a *Ret-RNAi* transgene from the adult progenitor driver *escargot* (*esg*)-*Gal4* (Fig 2A, C and F) and confirmed by Ret immunostaining (Fig EV1B). In parallel, we also analysed a newly generated *Ret* knock-out allele (confirmed by immunostaining; Fig EV1C, see Materials and Methods for details), either in whole mutants (Fig 2D and G) or using MARCM clones (mosaic analysis with a repressible cell marker (Lee & Luo, 1999; Fig 2B). In both approaches, quantifications of mitotic figures (phospho-histone 3 (pH3)-positive cells, Fig 2C–E), progenitor number (Fig 2A) and clone size (Fig 2B) revealed that reduction or loss of *Ret* function impairs stem cell proliferation. Reduced stem cell proliferation was observed both during normal homeostasis (Fig 2A and B) and in response to epithelial damage [damage induced by dextran sodium sulphate (DSS; Amcheslavsky *et al*, 2009), Fig 2C and D]. *Ret* downregulation or mutation also reduced the epithelial hyperplasia observed during normal ageing (Biteau *et al*, 2008; Choi *et al*, 2008; Fig 2F and G). Additional cell type-specific quantifications revealed that this reduction in stem cell proliferation did not result from stem cell loss, nor was it accompanied by abnormal differentiation of their progeny. Indeed, both the number of stem cells and the ratios of differentiated cell types resulting from *Ret* mutation were comparable to those observed in wild-type flies (Fig EV1A, B and D).

Having established that *Ret* is required to sustain stem cell proliferation, we sought to explore its sufficiency. pH3 quantifications indicated that misexpression of Ret, achieved by expression of a *UAS-Ret* transgene, either clonally or in all adult intestinal progenitors, resulted in increased mitoses (Fig 2E and H). Clone size quantifications revealed that the increased proliferation triggered by Ret misexpression gave rise to a larger number of progeny (Fig 2H).

Together, loss- and gain-of-function experiments indicate that adult intestinal stem cell proliferation is modulated by Ret; normal proliferation requires endogenous Ret levels, and can be further enhanced by increasing Ret expression cell-intrinsically above wild-type levels.

## A *Ret/wingless* positive feedback loop promotes stem cell proliferation in the adult *Drosophila* intestine

Several mitogenic signals have been shown to promote stem cell proliferation when mis- or over-expressed from adult intestinal progenitors (Jiang & Edgar, 2012; Guo *et al*, 2016; Jiang *et al*, 2016). To begin to explore the signalling context in which Ret may function, we tested the ability of such mitogens to stimulate proliferation of adult intestinal progenitors with reduced *Ret* expression. To this end, we expressed ligands for the Jak-Stat, Egfr and Wingless (Wg)/Wnt signalling pathways [unpaired 1 (Upd1), secreted Spitz (sSpitz) and Wg, respectively] together with a *Ret-RNAi* transgene from adult intestinal progenitors. *Ret* downregulation did not render stem cells unable to respond to all mitogenic signals, but selectively reduced *wg*-induced proliferation (as revealed by pH3 quantifications, Fig 3A). Co-expression of *Ret-RNAi* and a dominant-negative form of Shaggy (Sgg), the *Drosophila* orthologue of glycogen synthase kinase 3β (Gsk3b), indicated that *Ret* is also required for the increased proliferation resulting from activating Wg signalling in adult intestinal progenitors downstream of the Wg receptor (Fig 3A).

We then explored the interaction between these two pathways in the converse scenario: Ret-induced proliferation. We found that *Ret* over-expression in adult intestinal progenitors leads to striking upregulation of Wg ligand in these cells (Fig 3B). Experiments in which we simultaneously over-expressed Ret and downregulated Wg signalling in adult intestinal progenitors [achieved by co-expression of *RNAi* transgenes the downstream Wg pathway component *dishevelled* (*dsh*), or by expression of a dominant-negative version of the transcription factor target pangolin/TCF (Pan)] indicated that *Ret*-driven proliferation requires active Wg signalling in adult intestinal progenitors (Fig 3C).

Although several studies have reported contributions of Wg signalling to intestinal stem cell homeostasis (Lin *et al*, 2008; Cordero *et al*, 2012; Tian *et al*, 2016), they have reached different conclusions regarding the source of Wg protein. To test whether the Ret-induced Wg upregulation results from cell-intrinsic Wg production by adult intestinal progenitors, we first quantified *wg* transcript by RT–qPCR in dissected midguts following over-expression of Ret in adult intestinal progenitors. In contrast to the *wg* transcript upregulation detected following *wg* expression in these cells (positive control), Ret over-expression did not result in detectable *wg* transcript induction (Fig 3D). This suggested that the increased Wg protein observed following Ret expression may result from a non-transcriptional mechanism. To confirm this, we analysed *wg*

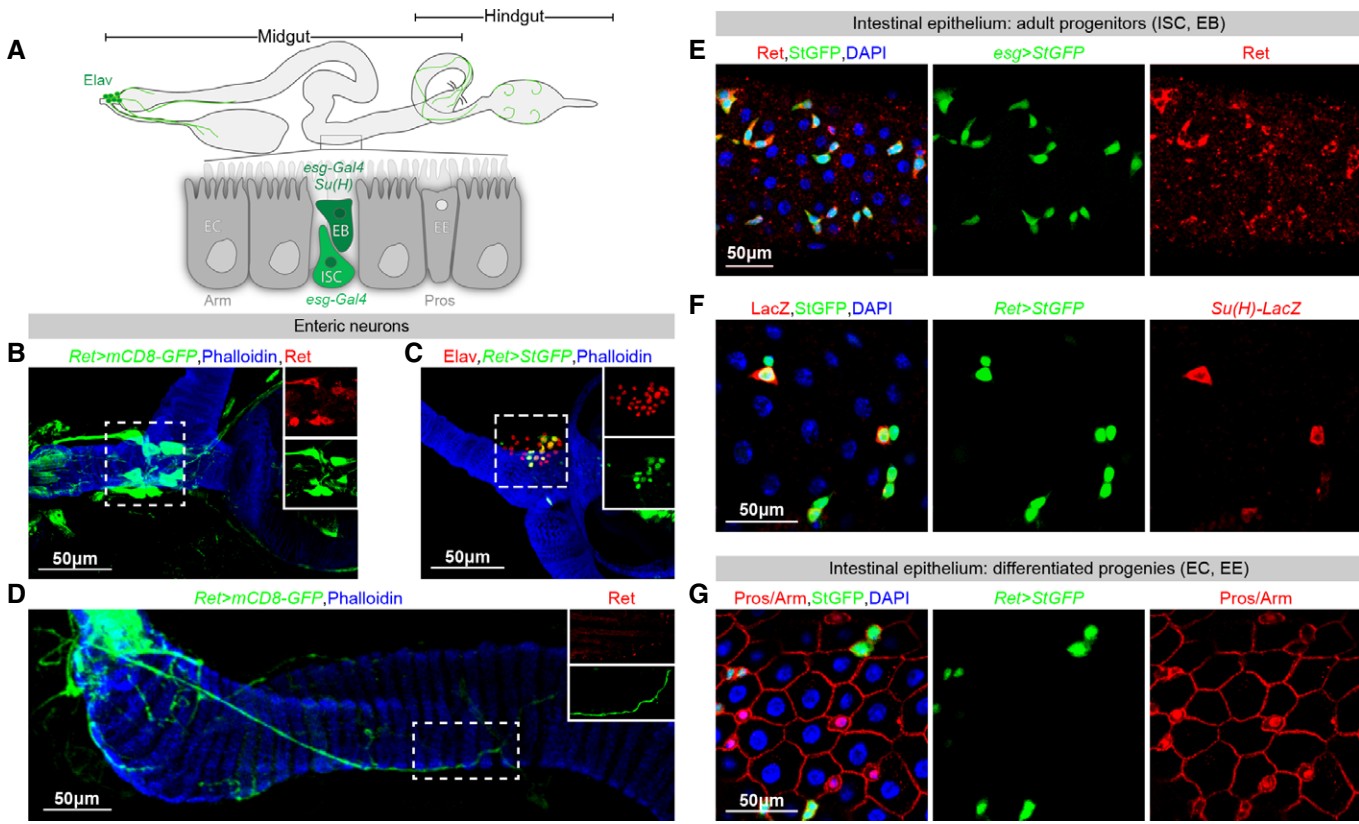

**Figure 1.  Ret is expressed in the *Drosophila* adult midgut.**

A   Cartoon summarising different cell types in the adult midgut and the immunohistochemical markers used to identify them. Ret-expressing cells are highlighted in green and include enteric neurons (the nuclei of which are embryonic lethal abnormal vision (Elav)-positive) and two types of adult intestinal progenitors: stem cells (ISCs, *escargot* (*esg*)-positive, *Suppressor of Hairless* (*Su(H)*)-negative) and enteroblasts (EBs, *esg*- and *Su(H)*-positive). Ret is absent from visceral muscle (Phalloidin-positive) and from most of  the two other epithelial cell types: enteroendocrine cells (EEs, Prospero (Pros)-positive) and enterocytes (ECs, labelled as Armadillo (Arm)-positive cells with large endoreplicating nuclei, visualised with DAPI).

B   Expression of Ret in axons [labelled with a Ret antibody in red and a *Ret-Gal4*-driven membrane-tagged (CD8) GFP reporter Flybow 1.1 (Hadjieconomou *et al*, 2011)] innervating the anterior midgut.

C   *Ret-Gal4*-driven expression of a nuclear GFP reporter [*Stinger GFP* (Barolo *et al*, 2000)] indicates that some of the anterior midgut innervation emanates from neuronal cell bodies (co-labelled with anti-Elav) located in one of the enteric ganglia: the hypocerebral ganglion.

D   Expression of Ret protein (in red) and the same reporter as in (B) in axons of hindgut-innervating neurons.

E   All Ret-positive epithelial cells co-express the ISC/EB marker *esg-Gal4*.

F   Co-staining of the *Ret-Gal4* reporter with a *Su(H)*-LacZ reporter indicates that the doublets of small Ret-positive cells contain one *Su(H)*-negative ISC and one Su(H)-positive EB.

G   Co-staining of the *Ret-Gal4* reporter with the cell membrane marker Arm and the EE nuclear marker Pros indicates that neither EEs (Arm, Pros[+]) nor ECs (Arm[+] cells with large DAPI nuclei) express Ret, although very low levels of Ret can be detected in a few ECs (data not shown).

Data information: In panels (E–G), DAPI is used to visualise all nuclei. For full genotypes, see the Appendix.

---

transcription with cell specificity by making use of *wg^KO-Gal4*: a *wg* transcription reporter in which Gal4 is expressed under the control of the endogenous *wg* promoter (Alexandre *et al*, 2014). As described in a previous study (Tian *et al*, 2016), we found no expression of this reporter in adult intestinal stem cells (Fig 3E), but were able to detect it at intestinal compartment boundaries (data not shown). Consistently, Ret expression from this Gal4 driver failed to promote intestinal stem cell proliferation (Fig 3E).

Together, these results indicate that a positive, reciprocal interaction between Ret and Wg pathways promotes intestinal stem cell proliferation. Ret expression in adult intestinal progenitors promotes *wg* signalling in these cells via a non-transcriptional mechanism that increases Wg protein on their surface.

**Ret-dependent phosphorylation of Src42A and Fak kinases affects proliferation of adult intestinal progenitors**

What cellular events downstream of Ret promote Wg signalling and intestinal stem cell proliferation? Previous studies have revealed a complex and sometimes contradictory interplay between phosphorylation events resulting from Ret receptor activation involving binding to/phosphorylation of Src and focal adhesion kinases (Melillo *et al*, 1999; Encinas *et al*, 2001, 2004; Sandilands *et al*, 2012). To explore the significance of these interactions in the context of intestinal homeostasis *in vivo*, we first tested whether phosphorylation of Ret receptor was required to promote intestinal proliferation. To this end, we generated transgenic flies expressing a

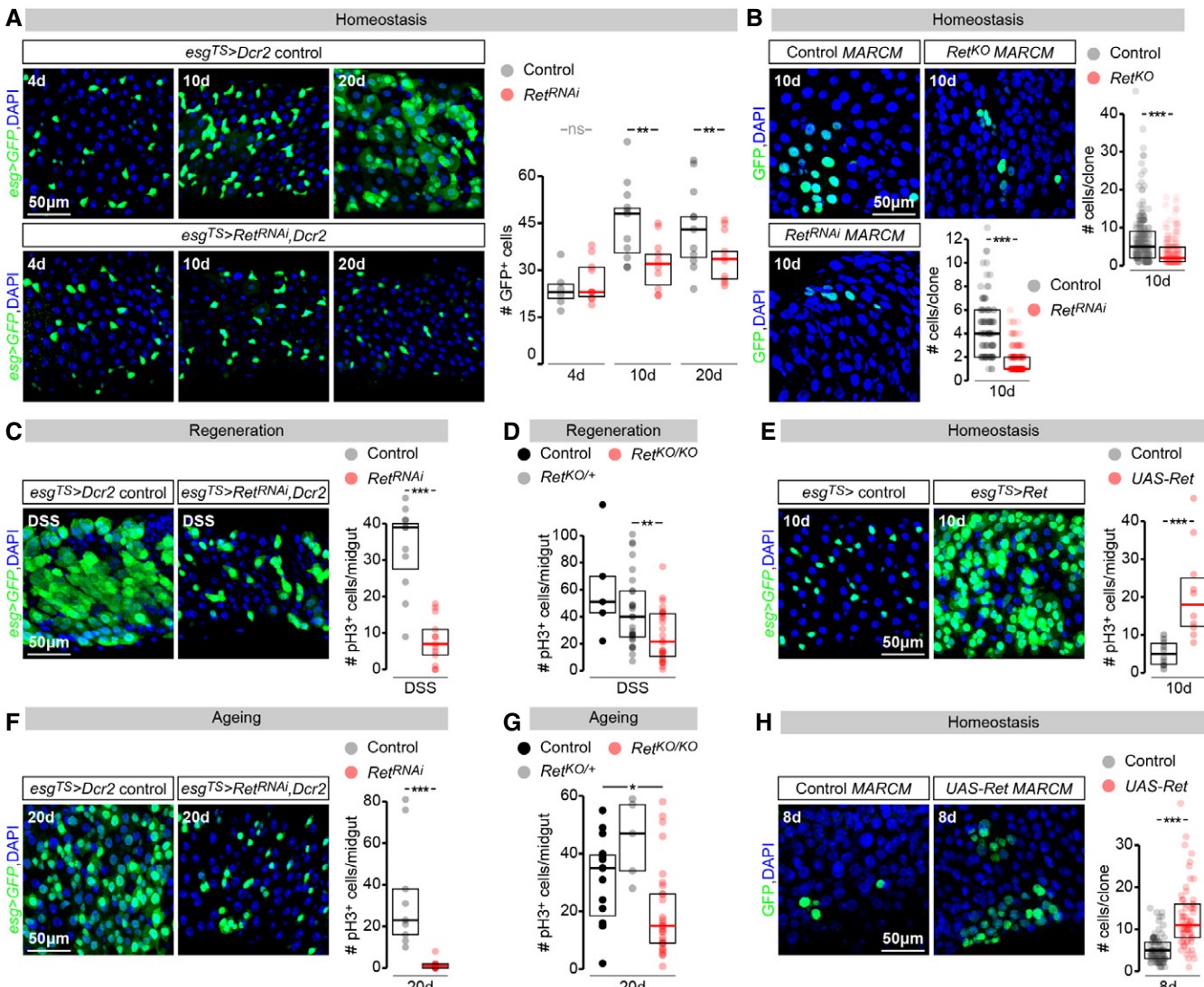

**Figure 2.  Ret levels modulate adult ISC proliferation.**

A   Representative images (left) and quantifications (right) of the number of intestinal progenitor cells in control midguts or midguts in which *Ret* has been downregulated from adult ISCs/EBs [achieved by *esg-Gal4*, *tub-Gal80^TS*-driven *Ret-RNAi*, enhanced by *UAS-Dicer2* (*Dcr2*) co-expression (Dietzl *et al*, 2007)] for 4, 10 or 20 days.

B   MARCM clone size quantifications (graph) and representative images (clones labelled in green with GFP) reveal that clones lacking *Ret* (*Ret^KO*) or expressing *Ret-RNAi* are smaller than control clones 10 days after clone induction.

C   Quantifications of mitoses (pH3-positive cells, graph) and visualisation of intestinal progenitors (using *esg*-driven GFP, image panels) in midguts of flies with the same genotypes as in (A). The regenerative response triggered by damage-inducing DSS in control flies is reduced following *Ret* downregulation from ISC/EBs.

D   pH3 quantifications of DSS-damaged midguts of wild-type control, *Ret* heterozygous (*Ret^KO/+*) and *Ret* mutant (*Ret^KO/KO*) flies.

E   Representative images of the number of intestinal progenitors (left) and pH3 quantifications (right) in control midguts or midguts in which *Ret* has been over-expressed from adult ISCs/EBs (achieved by *esg-Gal4*, *tub-Gal80^TS*-driven *Ret* misexpression) for 10 days. In both image panels, intestinal progenitors (ISC/EBs) are labelled with *esg-Gal4*-driven GFP.

F   Quantifications of mitoses (pH3-positive cells, graph) and visualisation of intestinal progenitors (using *esg*-driven GFP, image panels) in midguts of flies with the same genotypes as in (A), aged for 20 days.

G   Midgut pH3 quantifications of wild-type control, *Ret* heterozygous (*Ret^KO/+*) and *Ret* mutant (*Ret^KO/KO*) flies aged for 20 days.

H   MARCM clone size quantifications (graph) and representative images (clones labelled in green with GFP) reveal that clones over-expressing *Ret* (*UAS-Ret*) are larger than control clones 8 days after clone induction.

Data information: In all image panels, DAPI is used as a generic nuclear marker, and intestinal progenitors (ISC/EBs) are labelled with *esg-Gal4*-driven GFP. For full genotypes, see the Appendix. Values are presented as average ± standard error of the mean (SEM). *P*-values from Mann–Whitney–Wilcoxon test (ns, $P > 0.05$; *$P < 0.05$; **$P < 0.01$; ***$P < 0.001$).

kinase-dead Ret protein harbouring a substitution of Lys805 into Met (RetK805M): a mutation previously shown to abrogate the high levels of Tyr autophosphorylation observed following expression of

wild-type *Drosophila* Ret in COS cells (Abrescia *et al*, 2005; see Materials and Methods for details). In contrast to the mitogenic effects of wild-type Ret, RetK805M failed to promote proliferation of

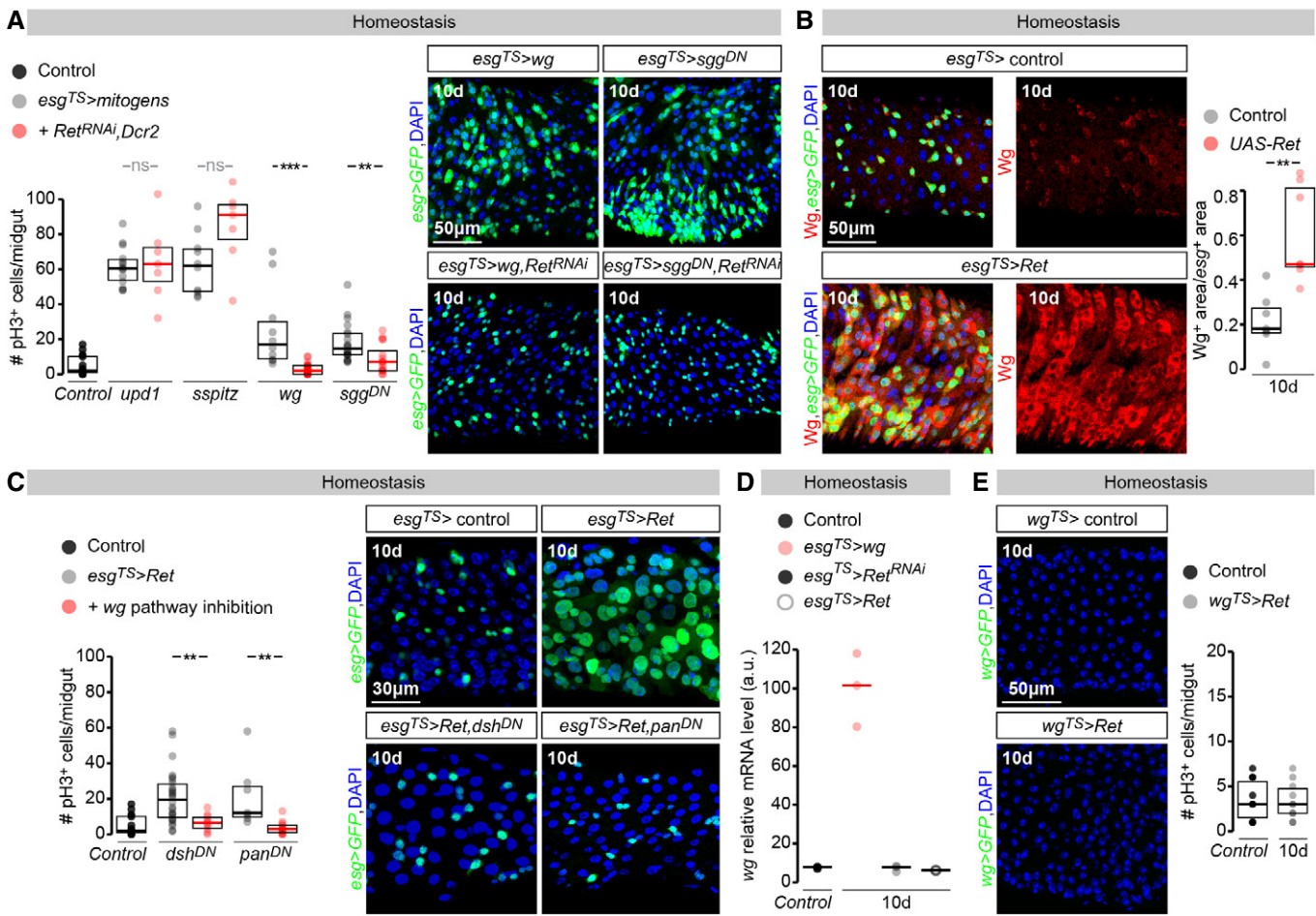

**Figure 3. A positive, autoregulatory Ret/Wg feedback loop sustains intestinal stem cell proliferation.**

A    pH3 quantifications indicate that *upd1*, *sspitz*, *wg* or *sgg^DN* expression in adult intestinal progenitors (achieved using *esg-Gal4*, *tub-Gal80^TS*) all promote their proliferation. Simultaneous downregulation of *Ret* (by co-expression of one of the above genes together with *Ret-RNAi* and *UAS-Dcr2*) significantly reduces the *wg-* or *sgg^DN*-triggered proliferation increase, but not that triggered by other mitogens. Image panels (right) show representative images of the number of intestinal progenitor cells (labelled with *esg-Gal4*-driven GFP) in midguts exposed to ectopic *wg* and *sgg^DN*, with and without co-downregulation of *Ret*.

B    Representative images of Wg ligand levels (left, labelled using an anti-Wg antibody) and Wg-positive area quantifications (right) in control midguts or midguts in which *Ret* has been over-expressed from adult ISCs/EBs (achieved by *esg-Gal4*, *tub-Gal80^TS*-driven expression of a *UAS-Ret* transgene) for 10 days. In both image panels, intestinal progenitors (ISC/EBs) are labelled with *esg-Gal4*-driven GFP.

C    pH3 quantifications indicate that the proliferation increase resulting from *Ret* over-expression in intestinal progenitors (achieved by *esg-Gal4*, *tub-Gal80^TS*-driven expression of a *UAS-Ret* transgene) is significantly reduced by co-expression of dominant-negative versions of Dsh or Pan (*dsh^DN* and *pan^DN*, respectively). Image panels (right) show representative images of the number of intestinal progenitor cells (labelled with *esg-Gal4*-driven GFP) in midguts in which *Ret* has been over-expressed in these cells, and its reduction when *Ret* is over-expressed together with *dsh^DN* or *pan^DN*.

D    *wg* transcript levels relative to *alphaTub84B* transcript levels in dissected midguts of control flies or flies expressing *wg*, *Ret* or *Ret-RNAi* from adult intestinal progenitors. Transcript levels were assessed 10 days after *Gal80* transgene induction.

E    Images to the left show lack of GFP expression from *wg^KO-Gal4* (abbreviated as *wg^TS>*) in the midgut epithelium, both in control homeostatic conditions (top) and upon expression of *Ret* in adult intestinal progenitors (bottom panel, 10 days after *Gal80* transgene induction). The right graph shows a quantification of the number of pH3-positive cells in the posterior midgut of flies following 10 days of adult-restricted expression of Ret from *wg^KO-Gal4* (abbreviated as *wg^TS>*) relative to control flies. The number of pH3-positive cells is not significantly different between the two groups of flies.

Data information: In all image panels, DAPI is used as generic nuclear marker. For full genotypes, see the Appendix. Values are presented as average ± standard error of the mean (SEM). *P*-values from Mann–Whitney–Wilcoxon test (ns, *P* > 0.05; **P* < 0.01; ****P* < 0.001).

adult intestinal progenitors, despite efficient expression of this mutated Ret protein in these cells (Fig 4A). This indicates that phosphorylation of Ret is required for its mitogenic effects in the intestine.

Several *ex vivo* studies have shown that, following its phosphorylation and activation, Ret can bind the non-receptor tyrosine kinase

Src (Melillo *et al*, 1999; Encinas *et al*, 2004). Consistent with this biochemical interaction, we observed that, in addition to upregulating Wg, Ret over-expression in adult intestinal progenitors also resulted in enhanced phospho-Src immunoreactivity (Fig 4B), suggestive of Src kinase activation. Activation of Src cytoplasmic tyrosine kinases is frequently implicated in signal transduction

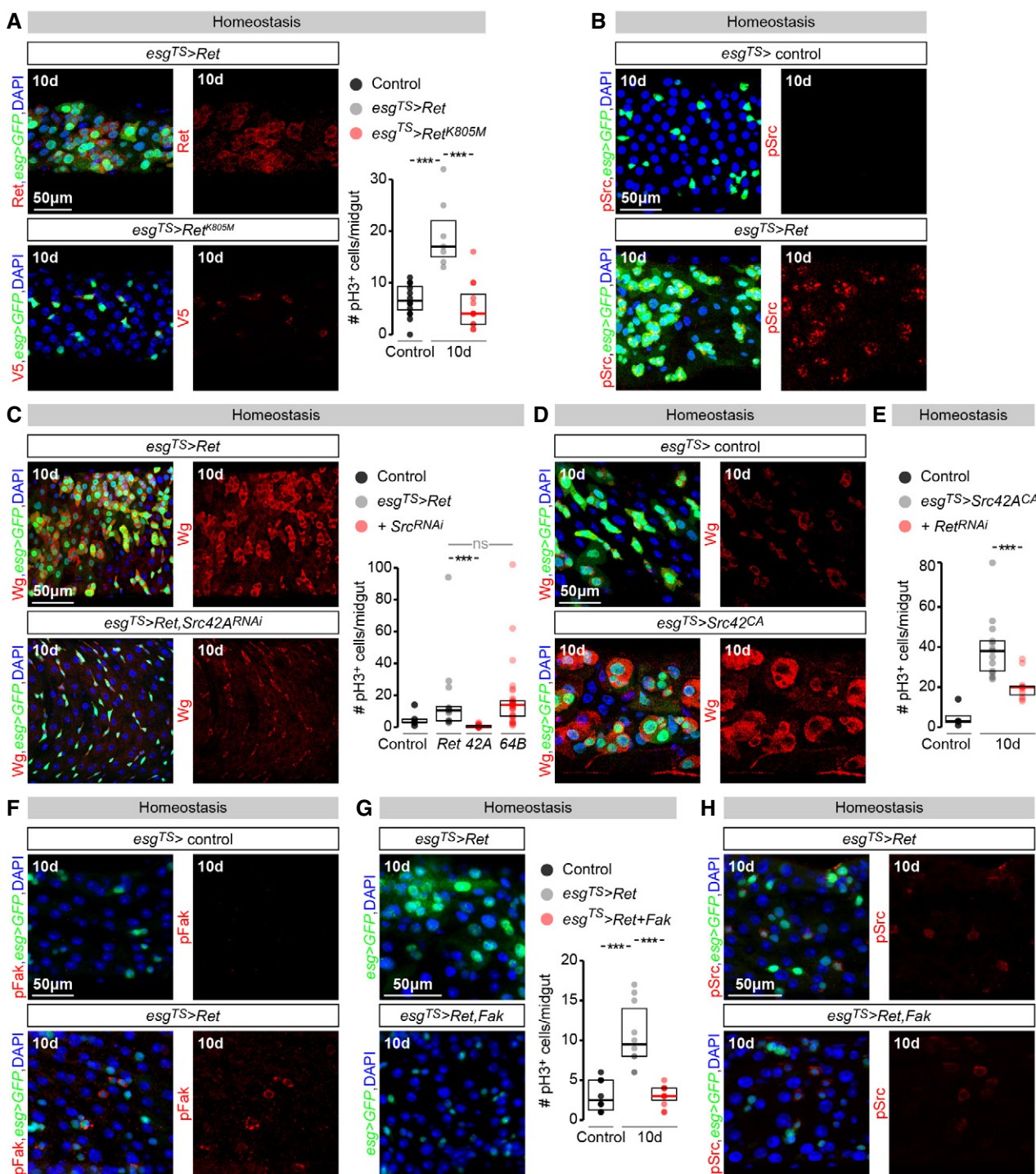

Figure 4.

downstream of transmembrane receptors, and modulation of Src levels in intestinal progenitors results in phenotypes similar to those described above for *Ret* depletion/over-expression (Cordero *et al*, 2014; Kohlmaier *et al*, 2015), suggesting that Ret-induced hyperproliferation may require Src kinases. Experiments in which we simultaneously over-expressed *Ret* and downregulated *Src42A* or *Src64B* (the two Src kinases in *Drosophila*) from adult intestinal progenitors

confirmed that *Src42A*, but not *Src64B*, is required for both *Ret*-driven hyperplasia and activation of Wg ligand in intestinal progenitors (Fig 4C). The existence of positive feedback between Ret and Wg pathways via Src42A was further suggested by the fact that activation of Src42A upregulated Wg ligand in intestinal progenitors (Fig 4D), and by our finding that the previously reported proliferation increase resulting from *Src42A* activation

◄

**Figure 4.  Ret-induced stem cell proliferation requires Src42A and is blocked by Fak.**

A  (Left) Representative images of the number of intestinal progenitors (labelled with *esg-Gal4*-driven GFP) and the efficiency of Ret misexpression (assessed by Ret (top) or V5 (bottom) immunostaining in red) in midguts in which *Ret* (control, top) or a V5-tagged, kinase-dead Ret (RetK805M, bottom) has been over-expressed from adult intestinal progenitors using *esg-Gal4, tub-Gal80$^{TS}$*. (Right) pH3 quantifications of the proliferation increase resulting from *Ret* over-expression in adult intestinal progenitors (control experiment), and its absence following over-expression of the kinase-dead Ret identical conditions.

B  Induction of pSrc (visualised using a pSrc antibody in red) resulting from *esg-Gal4, tub-Gal80$^{TS}$*- driven over-expression of *Ret* in adult intestinal progenitors (labelled with *esg-Gal4*-driven GFP).

C  (Left) Representative images of Wg ligand levels (labelled using an anti-Wg antibody in red) and the number of intestinal progenitors (labelled with *esg-Gal4*-driven GFP) in midguts in which *Ret* has been over-expressed from adult intestinal progenitors using *esg-Gal4, tub-Gal80$^{TS}$*, alone or together with a *Src42A-RNAi* transgene. (Right) pH3 quantifications of the proliferation increase resulting from *Ret* over-expression in adult intestinal progenitors, and its suppression by *Src42A*, but not *Src64B*, downregulation using *RNAi* transgenes.

D  Representative images of Wg ligand levels (visualised using an anti-Wg antibody) in control midguts or midguts in which a constitutively active form of *Src42A* has been over-expressed from adult ISCs/EBs (achieved by *esg-Gal4, tub-Gal80$^{TS}$*-driven *Src42A$^{CA}$* misexpression) for 10 days.

E  pH3 quantifications of ISC proliferation in midguts in which a constitutively active form of *Src42A* has been over-expressed from adult ISCs/EBs using *esg-Gal4, tub-Gal80$^{TS}$* for 10 days, alone or together with a *Ret-RNAi* transgene. *Ret* downregulation significantly reduces the proliferation increase resulting from *Src42A$^{CA}$* expression.

F  Induction of pFak (visualised using a pFak antibody in red) resulting from *esg-Gal4, tub-Gal80$^{TS}$*-driven over-expression of *Ret* in adult intestinal progenitors (labelled with *esg-Gal4*-driven GFP).

G  (Left) Representative images of the number of intestinal progenitors (labelled with *esg-Gal4*-driven GFP) in midguts in which *Ret* has been over-expressed from adult intestinal progenitors using *esg-Gal4, tub-Gal80$^{TS}$*, alone or together with Fak. (Right) pH3 quantifications of the proliferation increase resulting from *Ret* over-expression in adult intestinal progenitors, and its suppression by *Fak* co-expression.

H  Representative images of Src phosphorylation (labelled using an anti-pSrc antibody in red) and the number of intestinal progenitors (labelled with *esg-Gal4*-driven GFP) in midguts in which *Ret* has been over-expressed from adult intestinal progenitors using *esg-Gal4, tub-Gal80$^{TS}$*, alone or together with Fak.

Data information: In all image panels, DAPI is used as generic nuclear marker. For full genotypes, see the Appendix. Values are presented as average ± standard error of the mean (SEM). *P*-values from Mann–Whitney–Wilcoxon test (ns, $P > 0.05$; ***$P < 0.001$).

(Cordero *et al*, 2014; Kohlmaier *et al*, 2015) was, at least partially, *Ret*-dependent (Fig 4E).

Both Ret and Src kinases can phosphorylate Fak *in vitro* (Erpel & Courtneidge, 1995; Plaza-Menacho *et al*, 2011). In the context of gain-of-function Ret mutations, different studies have reached opposite conclusions regarding the ability of Fak to enhance or suppress Ret-driven neoplasia (Panta *et al*, 2004; Plaza-Menacho *et al*, 2011; Sandilands *et al*, 2012; Macagno *et al*, 2014). Concurrent with Wg induction and pSrc upregulation, Ret expression in adult intestinal progenitors led to a small but detectable induction of pFak (Fig 4F). To determine the effect of Fak on the Ret-dependent proliferation of adult intestinal stem cells, we tested whether Ret-induced proliferation in adult intestinal progenitors could be modulated by Fak. We found that Fak co-expression suppressed the proliferation increase resulting from Ret expression (Fig 4G). pSrc stainings revealed that the Src activation resulting from Ret expression was, however, unaffected by Fak (Fig 4H), indicating that Fak blocks the mitogenic effect of Ret downstream of Src activation.

Together, these experiments indicate that positive feedback signalling between Ret and Wg promotes proliferation in adult intestinal progenitors. The positive feedback between these two pathways requires Src42A kinase and can be blocked by Fak kinase.

### Ret is expressed in the mouse intestinal epithelium

Is the epithelial expression of Ret confined to the fly intestine? RT–qPCR expression analysis of Ret transcript in mice revealed strong Ret expression in all three portions of the mouse small intestine (duodenum, jejunum and ileum, Fig 5A). To determine whether some of the Ret transcript detected in these whole-tissue samples originated from the intestinal epithelium and to further investigate its developmental expression, we derived epithelial cultures from both developing and mature small intestinal tissue. Epithelial cells

from the adult small intestine of mice can be cultured *ex vivo* as budding organoids with distinct proliferative and differentiated domains (Sato *et al*, 2009). If epithelial cells are instead derived from the foetal epithelium or the proximal region of the neonatal intestine, they grow like spheroids (foetal enterospheres—FEnS) when subject to the same treatment. These FEnS lack differentiated lineages and consist of a seemingly homogenous proliferative cell population, seen as precursors to the adult stem cell population (Fordham *et al*, 2013). RT–qPCR analysis of epithelial cultures derived from embryonic or mature small intestine indicated that Ret transcript is present in both kinds of epithelial cultures (FEnS and organoids, respectively, Fig 3B). This suggested that the source of Ret transcript in the intestinal tissue preparations is not only enteric neurons/mesenchymal cells, but also epithelial cells.

To characterise the epithelial cell types contributing to Ret expression, we conducted *in situ* hybridisation and immunohistochemical analyses of small intestinal tissue. Scattered Ret-positive epithelial cells were consistently detected using both methods, both in immature small intestine (postnatal day 1 (P1) intestine) by *in situ* hybridisation (Fig 5C), and in mature epithelium by immunofluorescence (co-staining with epithelial marker E-cadherin in Fig 5D, specificity of epithelial signal validated in Fig EV2A and B). We were not able to detect Ret protein in the epithelium of immature intestines (data not shown), possibly because of relatively lower expression as suggested by the *ex vivo* cultures (Fig 5B).

Scattered Ret-positive cells were detected in the jejunum but were most prominent in the ileum (Fig 5D and E and data not shown). Their position within the epithelium is variable; we observed them at positions ranging from the top of the villi to the upper portion of crypts, but were typically located in the lower half of the villi (Figs 5D and E, and EV2A, C and D). Immunohistochemical analyses with cell type-specific markers failed to reveal an overlap with Paneth or goblet cell markers (lysozyme and mucin-2, respectively, Fig EV2C and D), but consistently indicated that most,

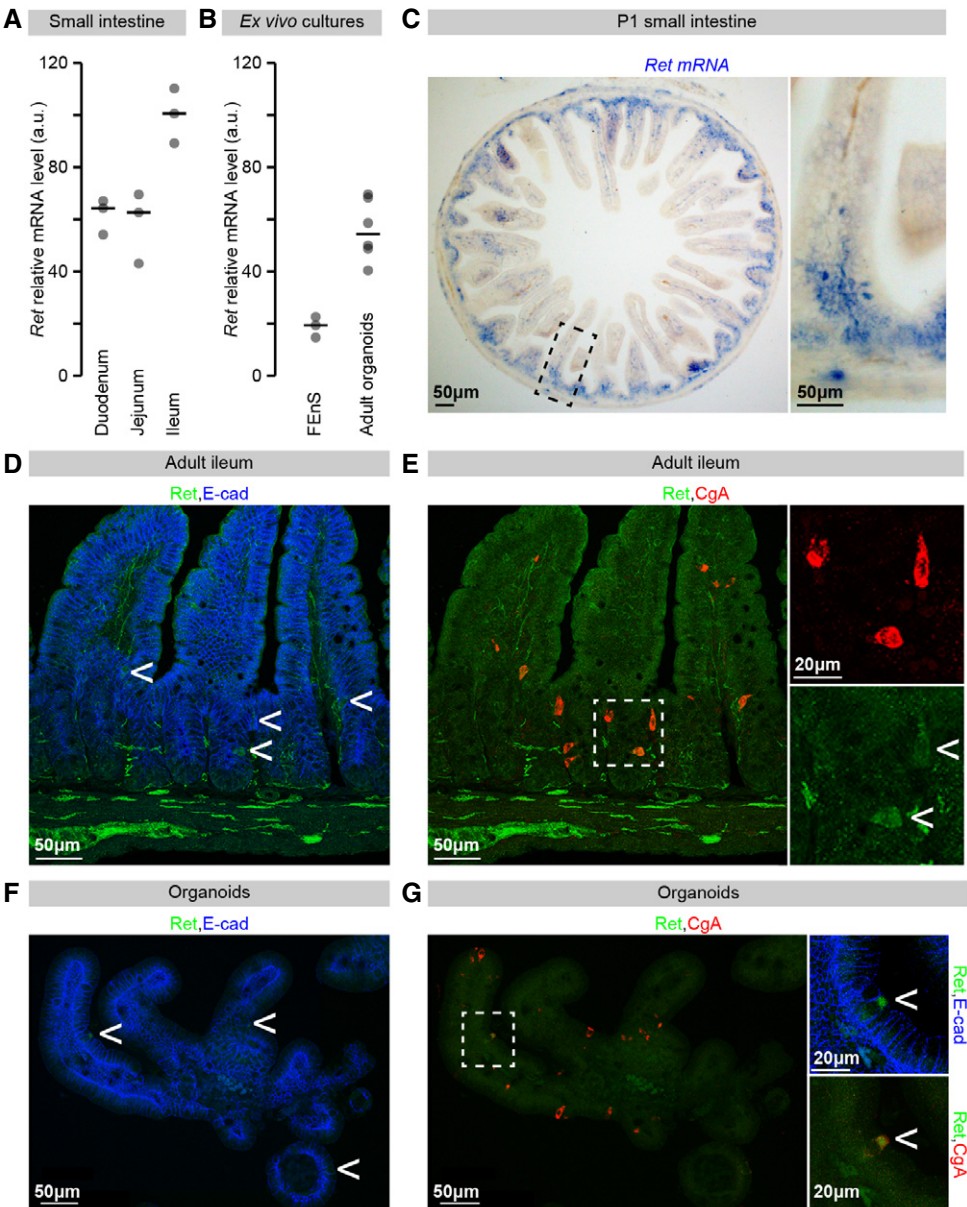

**Figure 5.  *Ret* expression in the mouse intestinal epithelium.**

A   Analysis of *Ret* transcript expression in adult small intestine by RT–qPCR. Ret transcript is detected in RNA samples from adult duodenum, jejunum and ileum.

B   *Ret* transcript is detected in RNA samples prepared from E16.5 FEnS cultures and organoid cultures derived from adult small intestine.

C   *In situ* hybridisation of *Ret* in P1 cross sections of small intestine. The higher magnification image to the right shows *Ret* transcript expression in scattered cells of a villus and intervillus.

D   Adult ileum sections co-stained with Ret (green) and E-cadherin (E-cad, in blue). In addition to the abundant neuronal fibres at the bottom of the image, Ret-positive cells can be observed amongst the epithelial cells, positive for E-cad (arrows).

E   Same section labelled with antibodies against Ret (green) and chromogranin-A (CgA, in red). Ret-positive cells are CgA-positive, but not all CgA-positive cells are Ret-positive.

F   Organoid cultures co-stained with Ret (green) and E-cad (blue). Ret- and E-cad-positive cells can be observed (arrows).

G   Single confocal section of organoid shown in (F) labelled with antibodies against Ret (green) and CgA (red). The Ret- and E-cad-positive cells are also CgA-positive.

Data information: Values are presented as averages, and each dot corresponds to an independent sample. In both (A and B), *Ret* transcript levels were normalised relative to *Actb* levels.

if not all, the Ret-positive epithelial cells correspond to a subset of chromogranin-A (CgA)-positive cells (Fig 5E), suggesting that Ret-positive cells are enteroendocrine in nature (Rindi *et al*, 2004; Bellono *et al*, 2017). Similarly scattered, Ret-, E-cadherin- and CgA-positive cells were also observed at equivalent positions in organoid cultures (Fig 5F and G).

Together, these data indicate that Ret is expressed in the developing intestinal epithelium. Ret expression is maintained in the mature intestinal epithelium in a subset of secretory, chromogranin-A-positive cells.

### Ret promotes maturation and Wnt signalling in the developing mouse epithelium

Having established that the mouse intestinal epithelium expresses Ret, we sought to assess the proliferation and behaviour of mouse epithelial cells with reduced Ret function. To this end, we used intestinal epithelial cultures derived from *Ret51* mice (de Graaff *et al*, 2001). We chose this mouse model because it lacks *Ret9*: one of two isoforms of the *Ret* gene, and the most abundant isoform in both whole intestinal tissue and epithelial organoids (Fig EV3A and B). Importantly, unlike $Ret^{-/-}$ mice that die perinatally, *Ret51* mice are viable for up to 7 days postnatally, allowing recovery of neonatal intestinal tissue. As previously reported, we observed that epithelial cultures derived from the proximal portion of neonatal (P1) small intestinal tissue of control mice comprise a mixed population of FEnS and organoids (Fordham *et al*, 2013; Fig 6B and C). We first confirmed that total *Ret* mRNA expression was reduced to 20% of its wild-type levels in the epithelium of FEnS/organoids derived from *Ret51* mice (Fig 6A). We then quantified the proportion of *Ret51* organoids relative to FEnS and found that it was reduced when compare to that of cultures derived from control littermates (Fig 6B and C). The increased relative abundance of primitive enterospheres in primary cell cultures derived from *Ret51* mutant intestine was further reflected in their reduced branching relative to cultures derived from control tissue (Fig 6D).

Analyses of cells isolated from the primary cell cultures indicated that both the fraction of viable *Ret51* cells and their cell cycle state were comparable to controls (Fig EV4A–C). This pointed to defective maturation, rather than proliferation or viability, as an underlying mechanism accounting for the *Ret51* epithelial phenotype. Consistent with this idea, transcriptional analysis of FEnS/organoid cultures derived from control and *Ret51* mice using a panel of epithelial differentiation markers revealed that, at a time point where most differentiation markers were only marginally reduced in cultures derived from *Ret51* mice (Fig EV4D), there was a strong reduction in *Axin2* transcript levels in these *Ret51* cultures (Fig 6E). *Axin2* is a commonly used readout of Wnt pathway activity: a signalling pathway that sustains intestinal stem cell proliferation and promotes both differentiation of Paneth cells *in vivo* and the maturation of FEnS to organoids *ex vivo* (Jho *et al*, 2002; van Es *et al*, 2005; Fordham *et al*, 2013; Clevers *et al*, 2014). To explore a possible positive feedback between Ret and Wnt signalling further, we treated immature cultures derived from wild-type E16.5 small intestines, consisting exclusively of FEnS, with GDNF: a Ret ligand. Exposure of FEnS to this Ret ligand robustly increased *Axin2* levels and promoted branching: a hallmark of maturation (Fig 6F–H). We further established that the effects of Ret on epithelial maturation were not confined to the *ex vivo* FEnS/organoid system. Indeed, immunohistochemical analysis of the neonatal small intestinal tissue of *Ret51* mice revealed reduced expression of intervillar lysozyme: a Paneth cell marker and one of the first hallmarks of tissue maturation, first detected in intervillus regions within the first week postnatally (Fordham *et al*, 2013; Fig 6I).

Together, these findings indicate that Ret impacts the transition between the ever expanding foetal epithelium and adult tissue homeostasis. Like in flies, Ret signalling can promote Wnt signalling to sustain epithelial maturation.

## Discussion

### A new role for Ret in intestinal epithelia: positive feedback between Ret and Wg/Wnt signalling

Our findings in *Drosophila* indicate that Ret is expressed not only by enteric neurons, but also by the adult somatic stem cells of the intestinal epithelium. In contrast to known Ret functions in other progenitor cell types—for example, in spermatogonia or the hematopoietic system (Naughton *et al*, 2006; Hofmann, 2008; Fonseca-Pereira *et al*, 2014)—Ret is not required for the survival of adult somatic stem cells in the intestine, but sustains both their homeostatic and regenerative proliferative capacity. Our gain- and loss-of-function experiments point to the existence of positive feedback between Ret and Wg signalling. Despite abundant genetic evidence that Wg signalling promotes stem cell proliferation in flies (Lin *et al*, 2008; Cordero *et al*, 2012; Tian *et al*, 2016), the source of Wg has remained unclear. Using new, improved tools to visualise Wg expression (Alexandre *et al*, 2014), our findings lend further support to recent data (Tian *et al*, 2016) indicating that the source of Wg ligand is not the stem cells themselves, despite the striking Ret-driven upregulation of Wg on their surface. How might Ret signalling in adult intestinal progenitors lead to Wg protein upregulation in these cells without affecting its transcript? Two possible ways in which it might do so is by upregulating the expression of Wg receptor(s) on their surface, and/or by promoting signalling from stem cells to the Wg-producing cells at the intestinal boundaries and/or the visceral muscles (Buchon *et al*, 2013; Tian *et al*, 2016) to increase Wg release/trafficking.

Epithelial Ret is not a peculiarity of the fly intestine; we find that *Ret* is also expressed in the developing intestinal epithelium of mice, prior to the maturation of enteroendocrine or Lgr5-positive stem cells. Although our immunohistochemical analyses have not revealed a specific Ret-positive cell population at this stage, our *ex vivo* experiments using epithelial cultures devoid of enteric neuron or mesenchyme point to an intrinsic role for Ret at this stage in promoting epithelial maturation. The mechanism underlying the maturation-promoting effects of Ret may involve positive feedback between Ret and Wnt signalling similar to those we have found in flies. Indeed, the Wnt pathway target *Axin2* is reduced in epithelial cultures derived from *Ret51* mice and upregulated when wild-type FEnS are treated with the Ret ligand GDNF: a treatment that also promotes their branching. These data are consistent with the previous finding that elevated Wnt signalling promotes FEnS to organoid maturation (Fordham *et al*, 2013) and is reminiscent of the Wnt11/Ret autoregulatory loop promoting ureteric branching during kidney development (Majumdar *et al*, 2003). The relevant source of Wnt driving tissue maturation is currently unknown and is most likely not epithelial (Farin *et al*, 2012; Kabiri *et al*, 2014; San Roman *et al*, 2014; Aoki *et al*, 2016; Valenta *et al*, 2016). Our *Drosophila* finding that Wg ligand upregulation is not transcriptional underscores the importance of considering tissue crosstalk

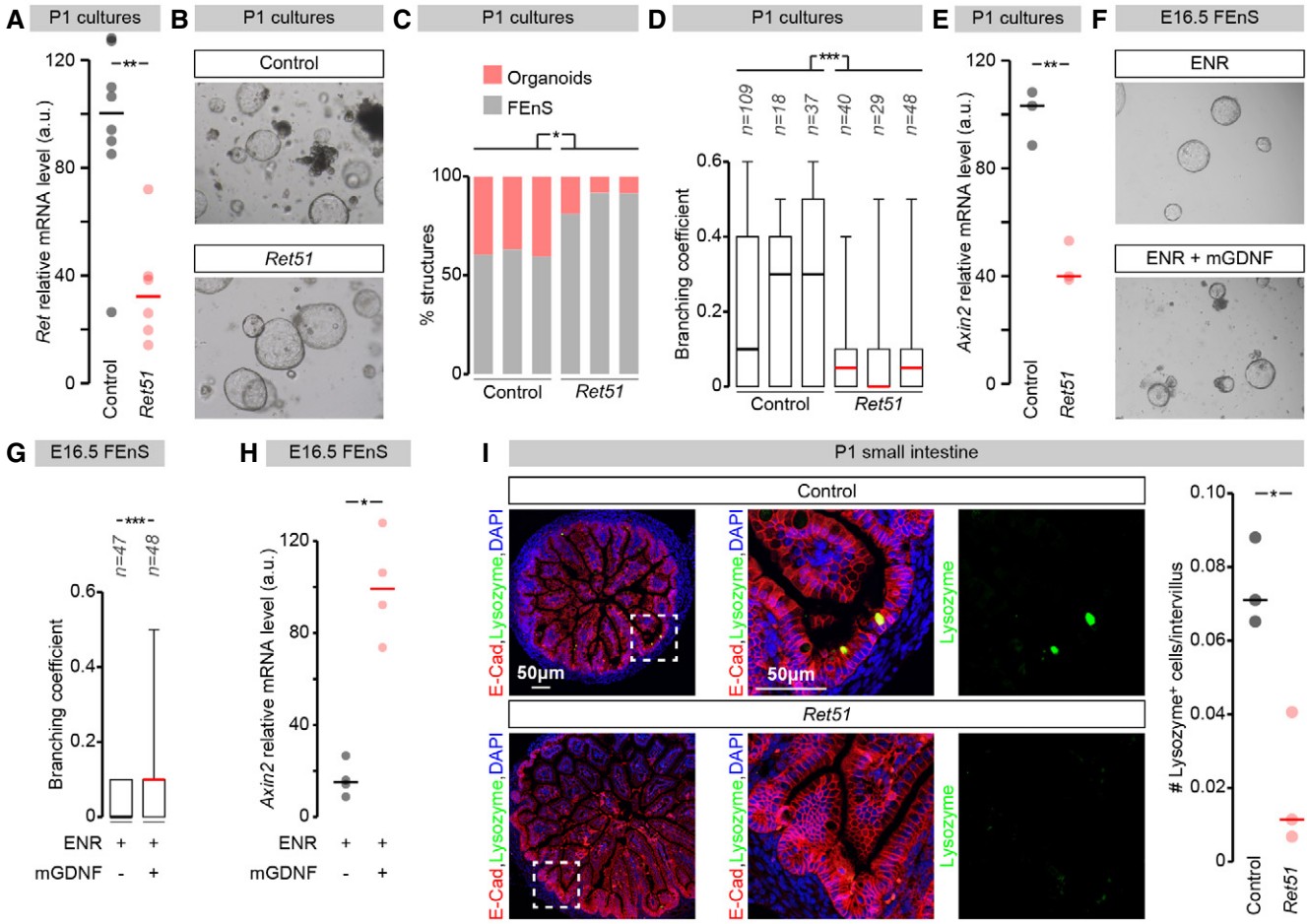

**Figure 6.  *Ret* control of epithelial maturation in mouse.**

A   *Ret* transcript levels relative to *Actb* transcript levels in RNA prepared from epithelial cultures derived from neonatal small intestinal tissue of *Ret51* mice or their control littermates.

B   Representative images of FEnS/organoid cultures derived from neonatal *Ret51* mice and their control littermates.

C   Quantifications of FEnS/mini-gut number in organoid cultures at passage 2. A higher proportion of FEnS is apparent in organoid cultures derived from *Ret51* mice.

D   Branching quantifications of organoids derived from *Ret51* mice and control littermates (see Materials and Methods for details). Organoids derived from *Ret51* mice are significantly less branched.

E   *Axin2* transcript levels relative to *Gapdh* transcript levels in RNA prepared from epithelial cultures derived from neonatal small intestinal tissue of *Ret51* mice or their control littermates.

F   Representative images of FEnS cultures derived from wild-type E16.5 mice, grown in control medium (ENR, top) or control medium supplemented with GDNF (ENR + GDNF, bottom).

G   Branching quantifications of the GDNF supplementation experiments in (F).

H   *Axin2* transcript levels relative to *Gapdh* transcript levels in RNA prepared from FEnS epithelial cultures grown in control medium (ENR) or control medium supplemented with GDNF (ENR + GDNF).

I   Representative images of lysozyme expression in small intestinal tissue. Lysozyme expression in intervilli appears to be less abundant in *Ret51* mice. Panels to the right are close-ups of lysozyme-positive intervilli belonging to the left panels. The right graph shows quantification of the numbers of lysozyme-positive cells in intervillar regions. *Ret51* mice have significantly fewer lysozyme-positive cells.

Data information: Values are presented as average ± standard error of the mean (SEM), and each dot corresponds to an independent biological replicate (A, E, H) or an independent sample (I). *P*-values from Mann–Whitney–Wilcoxon test (*$P < 0.05$; **$P < 0.01$; ***$P < 0.001$).

and non-autonomous signalling in any future studies addressing Wnt contributions to epithelial maturation in mice.

At first sight, the Ret effects on developmental maturation in mice appear to be different from its homeostatic role in flies. However, we find that Ret continues to be expressed in the adult small intestine, where Ret expression is prominent in a subset of enteroendocrine cells positive for the secretory marker chromogranin-A.

Based on their position, these cells may correspond to enterochromaffin cells: intriguing cells that contribute 90% of the serotonin in circulation, control gastrointestinal motility and secretions and have recently been shown to be chemosensory (Roth & Gordon, 1990; Gershon, 2003; Rindi *et al*, 2004; Gershon & Tack, 2007; Bellono *et al*, 2017). A very recent study has blurred the distinction between enteroendocrine cells and their precursors by revealing expression

overlaps between markers of enteroendocrine precursor identity and differentiated fate (including chromogranin-A), and by suggesting that differentiated enteroendocrine cells can have stem cell-like properties (Yan *et al*, 2017). This is exciting because it suggests that, whilst the cellular classification of Ret-positive cells based on known markers may differ between flies (ISCs) and mice (enteroendocrine), Ret-enabled stem cell functionality may contribute to regeneration in both epithelia. Intriguingly, endocrine tumours derived from the small intestine, ileal carcinoids, secrete serotonin and have been reported to express high levels of *Ret* (Cunningham & Janson, 2011). Conditional deletion of *Ret* in adult intestinal epithelium will, in future, determine its contribution to enteroendocrine fate and will help establish possible enteroendocrine contributions to intestinal homeostasis and tumour formation.

## Signalling events downstream of Ret

Consistent with *ex vivo* transfection studies in mammalian cells, pointing to physical association between Ret and c-Src (Melillo *et al*, 1999; Encinas *et al*, 2004), we find that Src kinase *Src42A* is required downstream of *Ret* to activate Wg signalling. These findings therefore strengthen the link between two pathways previously known to control ISC proliferation in flies—Src and Wg (Lin *et al*, 2008; Cordero *et al*, 2012, 2014; Kohlmaier *et al*, 2015)—and may provide a physiological context for the previously reported, Src-dependent mitogenic effects of mutated, oncogenic Ret in the *Drosophila* developing retina (Read *et al*, 2005). The finding that, in the *Drosophila* intestine, Src kinases control expression of a mitogenic module consisting of String/Cdc25 and cyclin E (Kohlmaier *et al*, 2015) provides a simple link between Ret activation and its pro-proliferative effects. Overactive Src kinases do, however, lead to intestinal tumours (Kohlmaier *et al*, 2015) so mechanisms must be in place to limit the positive Ret/Wg feedback loop so that it sustains homeostatic proliferation, but does not result in tumour formation. Availability of Wg ligand may be an extrinsic mechanism, but the focal adhesion kinase Fak may provide a cell-intrinsic break downstream of Ret/Src activation. Consistent with this idea, Ret expression leads to both Src42A and Fak phosphorylation, but we find that the two kinases have opposing effects on proliferation: Src42A promotes proliferation downstream of Ret, whereas Fak blocks it. Hence, despite the fact that blocking Fak function may represent a therapeutic opportunity in some cancers (Ashton *et al*, 2010; Lee *et al*, 2015), our findings are more aligned with a previous study (Macagno *et al*, 2014) that suggested that, at least in the context of Ret-driven tumorigenesis, Fak can act as a tumour suppressor. In future, it will also be of interest to explore how the Ras/Raf/Erk pathway, activated by Ret in other contexts and previously shown to affect ISC proliferation in flies (Buchon *et al*, 2010; Biteau & Jasper, 2011; Jiang *et al*, 2011), intersects with Src/Fak/Wg signalling in response to Ret activation.

Both Wnt and Src pathways can have strong effects on proliferation, differentiation and/or tumorigenesis in the murine intestine (Reya & Clevers, 2005; Jiang & Edgar, 2012; Cordero *et al*, 2014). Src is required for mouse intestinal tumourigenesis following upregulation of Wnt signalling (Cordero *et al*, 2014). Based on our functional findings in *Drosophila* and our expression data in mice, a possible contribution of the Ret- and CgA-positive cells to this process deserves further investigation. It will also be of interest to investigate how Src/Fak kinase signalling contributes to the maturation of foetal intestinal epithelial cells and whether this is important in the development of intestinal disorders.

## Epithelial Ret signalling in intestinal homeostasis and pathology

Ret expression is one of the defining features of enteric neurons (Sasselli *et al*, 2012). We have found another evolutionarily conserved and physiologically significant site of Ret expression: the intestinal epithelium. The presence of Ret in these two gastrointestinal cell types of very different developmental origin raises the possibility that the development and/or physiology of enteric neurons, intestinal epithelial progenitors and, in mammals, Ret-expressing intestinal lymphoid cells (Ibiza *et al*, 2016) is coordinated. Such coordination may, for example, help ensure a match between the size of the intestinal epithelium, the number of innervating neurons during development and the transition from an immature foetal epithelium into a functional epithelium involved in nutrient uptake and interorgan signalling. In mammals, Ret ligands of the glial cell line derived neurotrophic factor (Gdnf) family may orchestrate Ret signalling in these three tissues (Sasselli *et al*, 2012; Ibiza *et al*, 2016). In flies (which lack these Ret ligands), integrins have been shown to interact with Ret in sensory neurons (Soba *et al*, 2015). Interfering with integrin expression in the *Drosophila* intestine can have different effects on intestinal progenitor proliferation, survival and/or orientation depending on whether the integrins are removed from the progenitors or their niche—the visceral muscles (Goulas *et al*, 2012; Lin *et al*, 2013). Intriguingly, integrin downregulation in adult intestinal progenitors reduces their normal proliferation and can suppress their overproliferation in response to overactive Wingless signalling (Lin *et al*, 2013; and our own observations): phenotypes strikingly similar to those resulting from *Ret* downregulation. In the light of the known links between integrins and Fak/Src signalling in both normal and cancer cells (Mitra & Schlaepfer, 2006) and the effects that we have found for Src and Fak downstream of Ret activation, Ret could provide a new route for the integrin activation of the Src/Fak complex. Alternatively, the recent finding that GDF15, a divergent member of the TGF-β superfamily, signals through a GDNF family receptor α-like in a Ret-dependent way (Yang *et al*, 2017; Mullican *et al*, 2017; Emmerson *et al*, 2017) also raises the intriguing possibility that TGF-β-like ligands modulate Ret signalling in the intestinal epithelium, potentially linking intestinal regeneration with the known GDF15 roles in food intake/body weight.

The crucial requirement for Ret in enteric nervous system development is underscored by disorders such as HSCR, in which Ret loss of function leads to almost complete absence of enteric innervation in varying lengths of the distal gut (Schuchardt *et al*, 1994; Sasselli *et al*, 2012). Whilst the contribution of enteric aganglionosis to HSCR is unquestionable, our findings raise the possibility that, if the epithelial expression of Ret is conserved in humans, dysregulation of epithelial signalling may contribute to disorders that, like HSCR, result from *Ret* mutation. Epithelial Ret signalling might also contribute to other aspects of gastrointestinal physiology previously shown to be affected by reduced Ret function, such as intestinal motility, gut–microbiota interactions and the compensatory response to massive small bowel resection (Gianino *et al*, 2003; Hitch *et al*, 2012; Dey *et al*, 2015). Interestingly, HSCR is typically

diagnosed around birth due to defects in gastrointestinal functions. This coincides with the first demands on intestinal function, which could reflect not only neuronal defects related to peristalsis, but also defects associated with the transition from a foetal into a functional adult epithelium. Given that many of the pathways that drive tissue expansion and the maintenance of non-differentiated progenitor populations during foetal development are deregulated in cancer (Hu & Shivdasani, 2005), a possible contribution of Ret signalling to colorectal tumours also deserves further investigation.

# Materials and Methods

### Fly husbandry

Fly stocks were reared on a standard cornmeal/agar diet (6.65% cornmeal, 7.15% dextrose, 5% yeast, 0.66% agar supplemented with 2.2% nipagin and 3.4 ml/l propionic acid). All experimental flies were kept in incubators at 25°C, 65% humidity and on a 12-h light/dark cycle, except for those containing *tub-Gal80$^{TS}$* transgenes, which were set up at 18°C (restrictive temperature) and transferred to 29°C (permissive temperature) at the time when *Gal4* induction was required. This depended on the specific experimental requirements but typically, for loss-of-function (RNAi) or gain-of-function (UAS) experiments, flies were raised and aged as adults for 4 days at 18°C, were then shifted to 29°C to induce transgene expression, and adult midguts were dissected after 10–20 days (as indicated in each figure panel). Flies were transferred to fresh vials every 3 days, and fly density was kept to a maximum of 15 flies per vial. Only virgin females were used for all experiments.

For mutant ISC clonal analyses (MARCM clones), 4-day-old adults (raised and aged at 25°C) were heat-shocked for 1 h at 37°C to induce clones and were then kept at 29°C until dissection (8 days or 10 days thereafter as indicated in each figure panel). Flies were transferred to fresh vials every 3 days.

For damage-induced regeneration assays using *RNAi* transgenes, 4-day-old adult virgin flies were collected and raised at 18°C and were then shifted to 29°C for 10 days on standard media. Flies were transferred to fresh vials every 3 days. Flies were then transferred in an empty vial containing a piece of 3.75 cm × 2.5 cm paper. 500 ml of 5% sucrose solution (control) or 5% sucrose + 3 dextran sulphate sodium (DSS) solution was used to wet the paper, used as feeding substrate. Flies were transferred to new vial with fresh feeding paper every day for 3 days prior to dissection.

Damage-induced regeneration assays using mutant animals were conducted in an almost identical manner, but 4- to 10-day-old adult virgin flies reared at 25°C were used instead.

### Fly stocks

#### UAS transgenes
*UAS-Flybow 1.1* [Bloomington *Drosophila* Stock Centre (BDSC): 35537, (Hadjieconomou *et al*, 2011)], *UAS-StGFP* (*UAS-Stinger*, (Barolo *et al*, 2000), gift from M. Landgraf), *UAS-Dcr2* (*UAS-Dicer2*, (Dietzl *et al*, 2007), gift from I. Salecker), *UAS-Ret* [*UAS-Ret-3xFLAG-6xHis* (Kallijarvi *et al*, 2012)]; *UAS-upd1* (Jiang *et al*, 2009; gift from J. Cordero); *UAS-sspitz* [Schweitzer *et al*, 1995; gift from J. Treisman); *UAS-wg* (*UAS-wg::HA*, BDSC:5918); *UAS-sgg$^{DN}$*

(*UAS-sgg.A81T*, BDSC: 5360, (Bourouis, 2002)]; *UAS-dsh$^{DN}$* (*UAS-dshΔB21-2*, (Axelrod *et al*, 1998), gift from J. Axelrod); *UAS-pan$^{DN}$* (van de Wetering *et al*, 1997; gift from F.J. Díaz-Benjumea); *UAS-Src42A$^{CA}$* (BDSC: 6410); *UAS-Dcr2* (VDRC Stock Centre: 60007); *UAS-Ret-RNAi* (VDRC: GD 843); *UAS-dsh-RNAi* (VDRC: KK 101525); *UAS-Src42A-RNAi* (VDRC:KK 100708); *UAS-Src64B-RNAi* (VDRC: GD 35252), *UAS-Fak* (Palmer *et al*, 1999).

#### Reporters and Gal4 drivers
*Su(H)-LacZ* [*Su(H)$^{GBE}$-LacZ*, (Furriols & Bray, 2001)], *esg-Gal4$^{NP7397}$*, *UAS-GFP, tub-Gal80$^{TS}$* chromosome (gift from J. de Navascués), *Ret-Gal4* (Walker *et al*, 2013), *wg$^{KO}$-Gal4* (Alexandre *et al*, 2014; gift from Luis-Alberto Baena-Lopez).

#### MARCM stocks
FRT40A: *w, hs-flp, tub-Gal4, UAS-GFP; tub-Gal80, FRT40A* (gift from J. de Navascués).

### Generation of a *Ret* knock-out allele and a kinase-dead Ret

A targeted knock-out allele of *Ret* (*Ret$^{KO}$*) was generated using ends-out genomic engineering (Gong & Golic, 2003; Huang *et al*, 2009). Briefly, the *Ret* gene was replaced by a targeting cassette containing 5 kb and 3 kb of genomic sequences flanking the 5′ and 3′ region of *Ret* exons 3–8, respectively. *Ret$^{KO}$* candidate alleles were identified by correct chromosomal insertion of the targeting cassette containing a GMR-white marker cassette (pRK2; Huang *et al*, 2009). Candidates were analysed by PCR amplification and sequencing of genomic 5′ and 3′ regions flanking the targeting cassette to validate correct gene replacement. *Ret$^{KO}$* animals were viable and absence of Ret protein was further validated by immunohistochemistry of larval body wall preparations. A single fully validated *Ret$^{KO}$* allele was used for all subsequent experiments.

To generate flies expressing a kinase-dead version of Ret, we generated a Ret coding sequence harbouring a mutation resulting in a Lys to Met substitution at position 805 of the protein sequence. We fused this mutant Ret coding sequence with a sequence coding for a V5 tag and cloned the fusion into a pUAST vector.

Detailed description and analysis of both reagents will be provided elsewhere (N. Hoyer, P. Zielke, M. Petersen, K. Sauter, C. Hu, R. Scharrenberg, Y. Peng, C. Han, J.Z. Parrish, P. Soba, manuscript in preparation).

### Cell, clone and proliferation quantifications in adult *Drosophila* midguts

For quantifications of the *number of intestinal progenitors* (e.g. Fig 2A), confocal images were obtained with a Leica TCS SP5 upright confocal microscope using a 40× oil immersion objective. The images (1,024 × 512) were loaded into ImageJ, and they were reconstructed using maximum projection from the first three Z stack planes. These maximum projections were cropped into a new file of defined dimensions (width: 600 pixels, height: 150 pixels) to control for size differences, and the number of GFP-positive cells was then analysed (ImageJ function: Plug-ins>Analyze>Cell counter).

The *number of cells in MARCM clones* was quantified using a nuclear reporter. Images were acquired using a 63× oil immersion

objective. Z-stacks of four adjacent, but non-overlapping segments of the midgut were imaged, starting at the midgut-junction hindgut and moving anteriorly, and the number of cells per clone was counted in all clones found in these stacks (ImageJ function: Plugins>Analyze>Cell counter).

*Mitotic indices* were quantified by counting pH3-positive cells using a Nikon Eclipse 90i Fluorescence microscope through a 40× objective.

## Analysis of GFP and Wg levels

Confocal images were obtained with a Leica SP5 upright confocal microscope using a 40× oil immersion objective. A single stack (1,024 × 512) immediately posterior to the midgut–hindgut boundary was acquired with a Z resolution of 0.8 μm. ImageJ was used to generate a maximum projection using half of the Z stack planes. For GFP quantifications, the threshold was adjusted (ImageJ function: Image>Adjust>Threshold) to subtract background for the GFP channel and selected areas were then analysed (ImageJ function: analyse particles) to quantify the % of area above the threshold. To quantify Wg levels, the same procedure was applied for the red channel.

## Mouse strains and husbandry

*Ret51* mice, their control littermates or wild-type mice (C57BL/6j obtained from Charles River or C57BL/6NRj obtained from Janvier) were used as indicated in each experiment. Both male and female mice were used. All animals were placed under a 12-h light–dark cycle in an SPF unit in individually ventilated cages. Food and water were provided *ad libitum*. All mouse experiments performed in Copenhagen were approved by "Dyreforsøgstilsynet" (the national board approving animal experiments in Denmark). The experiments performed in the UK were conducted following guidelines from the National Centre for the Replacement, Refinement and Reduction of Animals in Research (NC3R) under project licenses granted by the UK Home Office Animals (Scientific Procedures) Act 1986.

## Immunohistochemistry

### Drosophila

Dissected *Drosophila* guts were fixed at room temperature for 20 min in PBS, 4% paraformaldehyde. All subsequent incubations were done in PBS, 4% horse serum, 0.2% Triton X-100 at 4°C following standard protocols.

The following primary antibodies were used at the following dilutions: guinea pig anti-Ret 1/1,000 (Soba *et al*, 2015), mouse anti-Pdm1 1/20 [kind gift from Steve Cohen, generated by Yeo *et al* (1995)], mouse anti-Wg 1/20 (4D4 Developmental Studies Hybridoma Bank), mouse anti-Arm 1/100 (N2 7A1 Developmental Studies Hybridoma Bank), mouse anti-Pros 1/50 (MR1A, Developmental Studies Hybridoma Bank), chicken anti-beta galactosidase 1/200 (ab9361, Abcam), goat anti-HRP 1/600 (123-095-021 Jackson ImmunoResearch), rabbit anti-p-Src 1/100 (44660G; Invitrogen), rabbit anti-pH3 (Ser10) 1/500 (9701L, Cell Signalling Technology), mouse anti-GFP 1/1,000 (11814460001, Roche), rabbit anti-V5 1/200 (V8137, Sigma) and rabbit anti-phospho-FAK (Tyr397) 1/250 (3283, Cell Signalling Technology).

Fluorescent secondary antibodies (FITC-, Cy3- and Cy5-conjugated) were obtained from Jackson Immunoresearch. Vectashield with DAPI (Vector Labs) was used to stain DNA.

### Mouse

P1 intestines were dissected from male and female *Ret51* mice, and littermate controls. Tissue was fixed overnight at 4°C in 4% paraformaldehyde and was subsequently embedded in paraffin. Fixed and embedded tissue was sectioned (5 μm) and was subsequently dewaxed. Then, the slides were processed for antigen retrieval by incubating them in Diva Decloaker (Biocare Medical) using 2100 Antigen Retriever. Following permeabilisation and blocking with 0.3% Triton X-100, 10% adult bovine serum and 5% skim milk powder, samples were incubated overnight at 4°C with the primary antibody. The next day, the slides were washed in PBS and incubated with secondary antibodies PBS 0.1% BSA for 1.5 h at room temperature. This was followed by washes with PBS and mounting with Vectashield-DAPI mounting medium.

Adult small intestines were dissected from male and female wild-type mice (C57BL/6) obtained from Charles River. Small intestines were fixed overnight at 4°C in 4% paraformaldehyde. They were then incubated in 30% sucrose solution and embedded in a mix of 7.5% gelatin (Sigma, G2500) and 15% sucrose in PBS until set. They were then snap-frozen in isopentane (Sigma, 154911) cooled at −60°C. Frozen samples were sectioned (14 μm thick) transversally using a Microm HM 560 CryoStar cryostat (Thermo Fisher Scientific). Sections were mounted onto SuperFrost Plus (Thermo Fisher Scientific) glass microscope slides. The sections were air-dried for 1–2 h at room temperature and were subsequently processed for antigen retrieval treatment by incubating them in sodium citrate buffer (10 mM sodium citrate, pH 6, Sigma-W302600) at 80°C for 15 min. Organoid cultures were fixed for 15 min with 4% paraformaldehyde a room temperature and processed whole mount. Following permeabilisation with 0.3% PBT and blocking in 10% horse serum, samples were incubated overnight at 4°C with the primary antibodies. The next day, they were incubated after a few PBT washed with appropriate fluorophore-conjugated secondary antibodies for 2 h at room temperature. This was followed by washes with PBT and mounting with Vectashield-DAPI mounting medium.

Antibodies used were as follows: Ret 1/300 (Immune Systems GT-15002), chromogranin-A 1/1,000 (Immunostar 20085), lysozyme 1/500 (Thermo Fisher Scientific PA1-29680), mucin 2 1/500 (Santa Cruz sc-15334) and/or E-cadherin ECCD-2 1/300 (Thermo Fisher Scientific 13-1900 (Invitrogen)). Alexa-488 and Alexa-555 (Thermo Fisher Scientific) were used as secondary antibodies.

## Image acquisition

Confocal images were obtained with a Leica TCS SP5 upright confocal microscope. Z-stacks planes were typically collected every 0.8 μm, and the images (1,024 × 1,024) were reconstructed using maximum projection. Confocal maximum projections were loaded into ImageJ, and level and channel adjustments were applied to the confocal images shown in the figures (the same correction was applied to all images belonging to the same experiment), but all quantitative analyses were carried out on unadjusted raw images or maximum projections.

## Statistics and data presentation

Sample size was not limiting in *Drosophila* experiments. Typically, 10–20 midguts were used for quantifications of midgut-level features such as the number of intestinal progenitors, expression of markers (e.g. Fig 1) or mitotic indices. An average of 30–40 clones was counted per midgut in experiments involving MARCM clones. In mouse experiments, power calculations to determine sample size were performed using expected changes in mean and standard deviation, available from the literature or from preliminary experiments conducted in our laboratories. A power of 0.8 was used in all power calculations. All experiments were performed using at least three independent biological replicates to allow for *post hoc* statistical analyses, and all animals were included in the analyses. For experiments involving organoids, lines were derived from independent animals, and treatment responses were normalised to the same line. Samples were not randomised, and the analyses were not blinded.

All statistical analyses were carried out in the R environment (R Core Team, 2014). Comparisons between two genotypes/conditions were analysed with the Mann–Whitney–Wilcoxon rank sum test (R function wilcox.test). All graphs were generated using Adobe Illustrator using the scatter graph tool. All confocal and bright field images belonging to the same experiment and displayed together in our figure were acquired using the exact same settings. For visualisation purposes, level and channel adjustments were applied using ImageJ to the confocal images shown in the figure panels (the same correction was applied to all images belonging to the same experiment), but all quantitative analyses were carried out on unadjusted raw images or maximum projections. In all figures, values are presented as average $\pm$ standard error of the mean (SEM), *P*-values from Mann–Whitney–Wilcoxon test (non-significant (ns): $P > 0.05$; *: $0.05 > P > 0.01$; **: $0.01 > P > 0.001$; ***$P < 0.001$).

## *In situ* hybridisation of mouse intestinal tissue

P1 intestines were fixed overnight at 4°C in 4% paraformaldehyde and were subsequently embedded in paraffin. Intestines were sectioned at 16 μm thickness. Digoxigenin-labelled antisense probes were generated from pBS SK plasmid with 2.5 kb of cDNA representing the entire intracellular and part of the extracellular domain of mouse *Ret* mRNA.

## Quantification of transcript composition and levels in mouse intestine/*ex vivo* cultures

*Ret* gene expression levels were analysed by reverse transcription–quantitative polymerase chain reaction (RT–qPCR). Total RNA was isolated from dissected intestines or intestinal cultures using Qiagen® MicroKit RNeasy. cDNA was synthesised using Invitrogen SuperScript III Reverse Transcriptase kit using random primers. For *Ret* expression in Fig 5A and B, TaqMan probes against a region common to both Ret isoforms were purchased from Thermo Fisher Scientific (*Ret* probe Mm00436304_m1 and *Actb* control Mm02619580_g1 for normalisation) and used together with a TaqMan Gene Expression Mastermix (Thermo Fisher Scientific). For *wg* expression in Fig 3D, the same procedure was used with TaqMan probes against *wg* (Dm01803389_g1) and *alphaTub84B*

(Dm02361072_s1) for normalisation. For all other RT–qPCR experiments, specific gene expression levels were measured using Invitrogen EXPRESS SYBR®GreenER™ qPCR Supermix with premixed Rox with optimised primer pairs in a Thermo Fisher Scientific QuantStudio™ 6 Flex Real-Time PCR System. Gene expression was normalised to *Gapdh* transcript levels. The primers used were as follows:

| Target | Forward sequence | Reverse sequence |
|---|---|---|
| GAPDH | 5′-TGTTCCTACCCCCAATGTGT-3′ | 5′-TGTGAGGGAGATGCTCAGTG-3′ |
| Lgr5 | 5′-CAGCGTCTTCACCTCCTACC-3′ | 5′-AGGAAGCAGAGGCGATGTAG-3′ |
| Fabp1 | 5′-TGGAAAGGAAGCCTCGTTGC-3′ | 5′-GCTTGACGACTGCCTTGACT-3′ |
| Muc2 | 5′-ACAAGCTGGCAGTGGTGAAC-3′ | 5′-AGACCTTGGTGTAGGCATCG-3′ |
| ChgA | 5′-GAACAGCCCCATGACAAAAG-3′ | 5′-TCGGAGATGACTTCCAGGAC-3′ |
| Ngn3 | 5′-GTTCCAATTCCACCCCACCT-3′ | 5′-GTTTGCTGAGTGCCAACTCG-3′ |
| Lyz1 | 5′-GCCAAGGTCTACAATCGTTGTGAGTTG-3′ | 5′-CAGTCAGCCAGCTTGACACCACG-3′ |
| Olfm4 | 5′-GCAGAAGGTGGGACTGTGTC-3′ | 5′-GCAGGGAAACAGAACACTGG-3′ |
| Ascl2 | 5′-CTACTCGTCGGAGGGAAG-3′ | 5′-ACTAGACAGCATGGGTAAG-3′ |
| Axin2 | 5′-TGAAACTGGAGCTGGAAAGC-3′ | 5′-AGAGGTGGTCGTCCAAAATG-3′ |

To determine the relative abundance of Ret isoforms, RNAseq reads from libraries originating from either adult small intestinal tissue (ileum of female mice) or adult organoids (derived from female whole small intestine) were aligned to mouse UCSC mm9 genome and gene models from Illumina's IGenomes using Tophat version 2.12. Relative transcript abundance Of Ret isoforms was calculated using Cufflinks version 2.11. For visualisation of isoform abundance, reads were allocated to transcripts using the GenomicAlignments Bioconductor package following the Cufflinks models and the assigned isoform signal visualised with the Gviz Bioconductor package.

## FEnS/organoid cultures

Proximal parts of small intestines (age as indicated for each experiment) of male and female *Ret51* mice, their littermate controls or wild-type animals (C57BL/6NRj obtained from Janvier) were isolated and then rinsed in PBS. The tissue was cut into 2–4 mm sections, transferred to clean PBS with 2 mM ethylenediaminetetraacetic acid (EDTA) and incubated by rocking it on ice for 45 min. The tissue was then re-suspended in clean PBS and dissociated by pipetting. Supernatant was collected and filtered through a 100- to 70-μm-pore-diameter nylon mesh. Crypts were seeded on 48-well plates in 25 μl of Corning® Matrigel® Growth Factor reduced (GFR) basement membrane mix, phenol red-free and ENR-Advanced DMEM/F1 (Gibco) (hEGF 50 ng/ml; Noggin 100 ng/ml, R-spondin1 500 ng/ml). The ENR medium was supplemented with murine GDNF 100 ng/ml (Peprotech) in the GDNF treatment experiments.

## FEnS/organoid branching quantifications

Branching coefficient was assessed from epithelial culture pictures using this formula in FIJI software:

$$\text{Branching coefficient} = 1 - \text{circularity} = 1 - \frac{4\pi \cdot \text{area}}{\text{perimeter}^2}.$$

## Cell cycle and cell death quantifications using dissociated cells

Single-cell suspensions from FEnS/organoids were obtained by incubation for 5–15 min at 37°C in 5 ml of Gibco® TrypLE™. For cell cycle analyses, cells were fixed in cold ethanol, washed in PBS and subsequently stained with DAPI. To detect apoptosis, single cell suspensions were stained using Annexin V Apoptosis detection Kit from BD Bioscience. Flow cytometry was performed using a BD® LSR2, and data were analysed using FlowJo® software.

**Expanded View** for this article is available online.

## Acknowledgements

We thank Yasuko Antoku (BRIC core Facility) and the Center for Advanced Bioimaging (CAB) Denmark, University of Copenhagen for technical assistance, and Julia Cordero and Vassilis Pachnis for helpful discussions. Susumu Hirabayashi provided comments on the manuscript. We are grateful to Jeff Axelrod, Luis Alberto Baena-Lopez, Sarah Bray, Steve Cohen, Joaquín de Navascués, Fernando Díaz-Benjumea, Matthias Landgraf, Bruno Lemaitre, Ruth Palmer, Iris Salecker and Jessica Treisman for providing reagents. This work was funded by an ERC Starting Grant and a Wellcome Trust Research Career Development fellowship to I.M.-A. (ERCStG 310411 and WT083559, respectively), a Lundbeck foundation fellowship to K.B.J. (R105-A9755), a Spanish Ministry of Education Fellowship to D.P., a Marie Curie Fellowship to J.G., an Academy of Finland Postdoctoral Fellowship to J.K. and MRC intramural funding. B.H. holds an EMBO advanced fellowship, and both K.B.J. and I.M.-A. were supported by the EMBO Young Investigator Programme.

## Author contributions

DP and IM-A designed research. DP, JG, BH, CK and AM performed the experiments, with contributions from DH and DN. TC analysed the RNAseq isoform data. NH, JK, JAW and PS generated fly reagents (transgenics/mutants). NT and AJB provided the *Ret51* mice. DP, BH, KBJ and IMA analysed and interpreted the data, with contributions from JG, CK and AM. IM-A wrote the manuscript.

## Conflict of interest

The authors declare that they have no conflict of interest.

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
