## [Review Process File · The EMBO Journal]

Manuscript EMBO-2016-96247

Ret receptor tyrosine kinase sustains proliferation and tissue maturation in intestinal epithelia

Daniel Perea, Jordi Guiu, Bruno Hudry, Chrysoula Konstantinidou, Alexandra Milona, Dafni Hadjieconomou, Thomas Carroll, Nina Hoyer, Dipa Natarajan, Jukka Kallijärvi, James A Walker, Peter Soba, Nikhil Thapar, Alan J Burns, Kim B Jensen, Irene Miguel-Aliaga

Corresponding author: Irene Miguel-Aliaga, Imperial College London

Review timeline:

Submission date:	05 December 2016
Editorial Decision:	12 February 2017
Revision received:	31 May 2017
Editorial Decision:	07 July 2017
Revision received:	26 July 2017
Accepted:	28 July 2017

Editor: Daniel Klimmeck

Transaction Report:

1st Editorial Decision

12 February 2017

Thank you for the submission of your manuscript (EMBOJ-2016-96247) to The EMBO Journal and sending us the preliminary point-by-point response to the referees' reports (which I enclose below). My apologies for the extended duration of the review process in this case. As outlined earlier, three referees have seen your manuscript, and they all highlight the potential interest and novelty of your study. However, at the same time the reviewers express reservations related to the mechanistic details on the uncovered novel role of Ret. Referee #1 states that the analogous function of enteric Ret in mouse and drosophila is not supported by your current data, and judges this part of the work as premature. This referee also asks you to work out the molecular function of Ret in intestinal homeostasis and development in greater detail. Referees #2 and #3 agree in that more analysis would be needed to address the molecular and physiological role of Ret in ISCs on both organisms, and that your claims on the autoregulatory loop are in their view not sufficiently supported by the data at this stage. These are important points in our view, which would need to be included in the revised manuscript. In summary, the referees do not see the novel insights sufficiently developed at this stage and ask you to strengthen the current conclusions by complementary and further reaching analyses.

In addition to the original reports, referee #1 has also seen your point-by-point response outlining experiments to be included in a revised manuscript. While the referee finds that your response would go a long way to address the referee concerns this person emphasizes the need for you to carefully explore the mechanism for the Ret/Src/Wnt autoregulation, which should in his/her view strongly improve the manuscript.

From my side, I judge the referee comments to be generally reasonable, thus we are in principle happy to invite you to submit a revised manuscript addressing the referees' comments.

REFeree REPORTS

Referee #1:

This paper reports a novel function for the Ret tyrosine kinase receptor in *Drosophila* intestinal stem cells, and the mouse intestinal epithelial cells. The authors show that, In *Drosophila*, Ret is required for and sufficient to drive ISC proliferation, and does so in part by activating Wgless expression and Src activity. A mechanistic analysis of how these effects are mediated is not provided. In the mouse intestine, the authors show that Ret is expressed in basal gut epithelium (where ISCs reside), and that guts lacking one of the two Ret genes exhibit a more fetal phenotype, lacking some characteristics of differentiation. The fly and mouse phenotypes are not readily attributed to analogous functions, and further studies would be required to determine how Ret effects its function in both systems. Overall the data provided is good quality and novel, but the study isn't extended far enough to yield significant new insights into the mechanisms of intestinal development and function. The study is very superficial. It would be more appropriate for EMBOJ if more specific, coherent conclusions could be offered relating to either Ret molecular functions or the roles of Ret in intestinal homeostasis and development.

specific comments:

1. Figure 3C requires a control expressing Ret (light grey control).
2. The activations of Wg expression and Src activity by Ret are very striking! Further information about the underlying mechanisms for this could add a lot to the paper's impact. Is the Wg induction seen for mRNA too? Is it cell autonomous? How is Src activated by Ret?
3. In Fig 4B, it is not possible to tell if the Ret mRNA is in ISCs or not. Which cell types express Ret?
4. Is Ret-regulated Wg expression conserved in mice? Can the mouse Wnts be upregulated by Ret in organoids? The same question can be asked for Src activity.
5. What is known about the mechanism of Ret signal transduction? What are the ligands in *Drosophila* and the mouse intestine? What are the downstream effectors? These questions should at least be discussed in more detail. Ideally, they would be addressed with data.

Referee #2:

Perea and colleagues report the characterization of expression and function of the Ret receptor in intestinal stem cells of flies and mice. Using immunohistochemistry, genetic reporters, genetic perturbations and lineage tracing, the authors show that Ret is expressed in ISCs and EBs in the fly gut and that it is required for ISC proliferation in both homeostatic and stress conditions. Ret is also sufficient to induce ISC proliferation in a Wg-dependent manner. Increased Ret expression results in increased Wg expression, as well as increased phosphorylation of Src42A. Src42A, in turn, is required and sufficient to induce Wg expression.

The authors go on to assess the expression of Ret in the mouse intestine and find that it is highly expressed in the epithelium. Organoids derived from Ret deficient animals show branching defects, and Ret deficient tissue suggest that Ret is required for maturation of the intestinal epithelium.

The study is interesting, as it introduces a new receptor influencing the wg pathway in intestinal stem cells. It is well-designed and executed, and the presented results are clear and properly interpreted. Given the importance of understanding the control of epithelial homeostasis for cancer

biology, it is clear that the study is of interest to a broad audience. There are a few concerns that the authors should address before publication can be recommended:

- The autoregulatory loop between Ret and Wg signaling is interesting, but also leads to difficulty in clearly establishing a mechanism for the role of Ret in ISCs. It may be useful to explore the activation of Wg expression and Src phosphorylation in a temporally resolved manner (for example in a time course after infection or DSS treatment). It can be anticipated that that would establish a temporal hierarchy of these events and may answer whether the interaction observed can be described as a linear pathway or whether there are parallel activation events of the two pathways that set up the autoregulatory pool.
- The question of why this autoregulatory loop is set up is not addressed. It seems necessary to propose a mechanism by which the autoactivation is temporally restricted (otherwise triggering this pathway would immediately result in dramatic over-proliferation). Experiments to address this question would be interesting.
- The analysis in the mouse needs improvement. More careful characterization of Ret expression in a spatially resolved manner would help (which cells actually express Ret?), and experiments that address the mechanism by which Ret influences maturation (does it influence Wg? Does it affect Paneth cell differentiation?) are necessary.

Referee #3:

This manuscript describes the role of the Ret receptor in intestinal stem cells of *Drosophila*. The results are novel, solidly demonstrated and of significant interest to the community.

I have only one minor comment, which is that the positive feedback loop between Wg-Arm and Ret-Src seems to lack any proposed mechanism. How does Ret-Src activate Wg expression? And how does Wg-Arm signalling activate Ret-Src? Some additional evidence along these lines would strengthen the manuscript.

1st Revision - authors' response

31 May 2017

Response to Reviewers Perea *et al.* Ret in intestinal epithelia

Referee #1

This paper reports a novel function for the Ret tyrosine kinase receptor in *Drosophila* intestinal stem cells, and the mouse intestinal epithelial cells. The authors show that, In *Drosophila*, Ret is required for and sufficient to drive ISC proliferation, and does so in part by activating Wingless expression and Src activity . A mechanistic analysis of how these effects are mediated is not provided. In the mouse intestine, the authors show that Ret is expressed in basal gut epithelium (where ISCs reside), and that guts lacking one of the two Ret genes exhibit a more fetal phenotype, lacking some characteristics of differentiation. The fly and mouse phenotypes are not readily attributed to analogous functions, and further studies would be required to determine how Ret effects it's function in both systems. Overall the data provided is good quality and novel, but the study isn't extended far enough to yield significant new insights into the mechanisms of intestinal development and function. The study is very superficial.

We are happy that the referee finds our data novel and of good quality. We have conducted additional experiments leading to new findings based on his/her suggestions – details are provided below in response to the specific points. However, we disagree that the study is “very superficial”, especially after the additional data provided in the revised manuscript. Our study has identified Ret as a new and important player in the regulation of intestinal stem cell homeostasis.

We show *in vivo* roles in normal homeostasis, regeneration and aging, and establish both its necessity and sufficiency. Using *Drosophila*-based genetic approaches comparable to those used in papers recently published by *EMBO J* or comparable journals (see, for example, PMID: 26077448, PMID: 27187149, PMID: 25433031, PMID: 25298397), we have established where and how Ret acts (uncoupling its neuronal from epithelial roles), and have revealed that

Ret is engaged in a positive feedback loop with Wingless/Wnt signalling, sustained by the Src kinase Src42A. This is an unusually specific feedback loop; our work has also shown that there is no such interaction between Ret and other pathways known to control stem cell proliferation. Furthermore, and unlike most *Drosophila* studies, we have provided mouse data. In the revised manuscript, we further shed mechanistic light on the Ret/Wg/Src feedback loop in both flies and mice (for example, by characterising the contributions of Ret autophosphorylation and Focal Adhesion Kinase to the process in flies and by showing that Ret and Wnt pathway regulate one another also in mice) and identify the epithelial cell population expressing Ret in the mouse intestine. Given that, for over 20 years now, all Ret gastrointestinal functions in health and disease have been attributed to the enteric nervous system, we feel that these are important findings.

It would be more appropriate for EMBOJ if more specific, coherent conclusions could be offered relating to either Ret molecular functions or the roles of Ret in intestinal homeostasis and development.

The new data has allowed us to draw more specific, coherent conclusions regarding the site, time and molecular nature of the Ret functions we described in the first version of the manuscript. We have included and discussed these in the revised manuscript. Details are provided in response to the specific points below.

specific comments:

1. Figure 3C requires a control expressing Ret (light grey control).

We were unsure which control(s) the referee alluded to. Controls expressing Ret have been provided for each experiment described in Figure 3C; light grey box plots to the left are all controls expressing Ret (all have the same *esgts>Ret* genotype, but each box plot corresponds to one set of controls for each experiment). Representative images of the *esgts>Ret* phenotype and *esgts>+* are also now provided to the right of the box plots to facilitate visual comparison to the genotypes in which *Ret* has been co-expressed with Wg pathway components. Genotypes for all controls are provided in the Full Genotypes table.

2. The activations of Wg expression and Src activity by Ret are very striking! Further information about the underlying mechanisms for this could add a lot to the paper's impact. Is the Wg induction seen for mRNA too?

We have tested this in two ways. We have quantified *wg* transcript levels using RT-qPCR following over-expression of Ret in adult-intestinal progenitors. In contrast to the *wg* transcript upregulation detected following *wg* expression in these cells (positive control), Ret over-expression does not result in a significant increase of transcript levels. The observed lack of transcript induction suggests that the increased Wg protein observed following Ret expression may result from a non-transcriptional mechanism. To confirm this, we analysed *wg* transcription with cell specificity by making use of *wgKO-Gal4*: a *wg* transcription reporter in which Gal4 is expressed under the control of the endogenous *wg* promoter (Alexandre *et al* 2014 PMID: 24390349). We find that this reporter is not expressed in adult intestinal stem cells. Consistently, Ret misexpression from this *Gal4* driver fails to promote intestinal stem cell proliferation. These results, which are now shown in Figures 3D and E, are consistent with recent findings by Buchon *et al* (PMID:23643535) and Tian *et al* (PMID:23643535), which strongly suggest that *wg* expression occurs at intestinal compartment boundaries rather than adult intestinal progenitors. Together, both the published expression data and our functional experiments make us favour a model whereby Ret signalling in adult intestinal progenitors results in increased binding of Wg protein to these cells, possibly through upregulation/stabilisation of Wg receptor(s). These considerations are now part of the revised discussion.

How is Src activated by Ret?

In mammalian cells, Src physically interacts with the Ret kinase domain. This is now clearly stated and referenced in the third (Ret/Wg) Results section. We have sought to extend these

findings in two ways. Firstly, *ex vivo* experiments have indicated that Ret/Src binding is known to require Ret autophosphorylation (PMID: 14766744 and 10070972). We have generated a kinase-dead version of the *Drosophila* Ret protein based on the findings of Abrescia et al 2005 (PMID: 15978587) and have tested its effects *in vivo* in adult intestinal progenitors. Unlike wild-type Ret, kinase-dead Ret is unable to promote adult intestinal stem cell proliferation, despite efficient expression in these cells. This indicates that Ret auto-phosphorylation is required for its mitogenic effects *in vivo*. These results are now shown in Figure 4A in the revised manuscript. Secondly, we have explored the interplay between Ret and Fak/Csk kinases: two non-receptor tyrosine-protein kinases that can physically interact with Ret and/or Src *ex vivo* (e.g. PMID: 18614016 and PMID: 21454698). Whilst our data so far has failed to implicate Csk in Ret signalling (for example, neither *Ret* gain- or loss-of-function in intestinal progenitors affected *Csk* levels, see Figure for Reviewers 1), we have found that Fak (shown to be a substrate of both Ret and Src kinases *ex vivo*, e.g. PMID: 21454698 and PMID: 8054685) is phosphorylated in adult intestinal progenitors in response to Ret expression. Unlike Src, however, Fak does not promote Ret-induced proliferation; in fact, its coexpression with Ret prevents the Ret-induced proliferation increase. Src is still phosphorylated in this condition, suggesting that Fak blocks the mitogenic effect of Ret downstream of Src activation. These results, now shown in Figures 4F-H, lead us to hypothesize that, following Ret auto-phosphorylation and Ret/Src activation, these two kinases may phosphorylate and, consequently, activate Fak. Fak, in turn, would provide a break that limits the proliferative effects of the positive Ret/Wg feedback loop. This is now discussed in the revised Discussion section.

3. In Fig 4B, it is not possible to tell if the Ret mRNA is in ISCs or not. Which cell types express Ret?

We have extended the mouse expression data in both mouse small intestinal tissue and organoids. In particular, we have:

- Extended the *Ret* transcript analysis by providing RT-qPCR data obtained from *ex vivo* epithelial cultures (FEnS and organoids) to confirm *Ret* transcript expression in intestinal epithelium. This is now shown in Figure 5B.
- Characterised the relative abundance of Ret isoforms in both adult tissue and organoids. In both samples, ca. 75% of transcripts code for the Ret9 isoform, further justifying the use of *Ret51* mice (lacking the Ret9 isoform) for our functional studies. This is now shown in Figure EV3.
- Immunostained both small intestinal tissue and organoids with an anti-Ret antibody. Costaining with an epithelial marker (E-cadherin) confirms that Ret is expressed in a subset of epithelial cells. This is now shown in Figures 5D and F (specificity of primary antibody shown in Figure EV2A and B).
- Co-stained both adult small intestinal tissue and organoids with Ret and cell-type specific markers (Paneth, goblet, enteroendocrine). This indicates that the subset of epithelial cells that express Ret are also positive for the enteroendocrine marker chromogranin-A. Costainings are now shown in Figures 5E, G and Figures EV2C and D. We have attempted similar stainings in FEnS and immature P1/P2 intestines, but have failed to detect a Ret antibody signal (Figure for Reviewers 2). We can, however, detect epithelial Ret transcript in immature epithelia, both by *in situ* hybridisation in neonatal intestine (Figure 5C) and RT-qPCR of epithelial FEnS cultures, where the levels may be lower than in organoids (Figure 5B). Hence, although we cannot identify a specific epithelial cell population expressing Ret at this stage (possibly because of lower expression), our transcriptional data points to epithelial *Ret* expression also in immature stages.

Together, this data confirms that Ret is expressed in mouse intestinal epithelium, and further indicates that Ret is expressed in a subset of secretory, chromogranin-A positive cells, present in both small intestinal tissue and organoids.

4. Is Ret-regulated Wg expression conserved in mice? Can the mouse Wnts be upregulated by Ret in organoids? The same question can be asked for Src activity.

We (the Jensen lab) have previously demonstrated that Wnt signalling drives intestinal maturation (PMID: 24139758). The observation that *Ret51* mutants display compromised

maturation is aligned with these findings. To extend these observations, we have explored how Ret signaling affects Wnt signalling in gain- and loss-of-function experiments. We have not used Wnts as readouts because, in mice, there are multiple sources of intestinal Wnt ligands including the mesenchyme and Paneth cells of the intestinal epithelium (see, for example, PMID: 22922422 and PMID: 21113151). The relevant source of Wnt driving tissue maturation, observed as the appearance of organoids and disappearance of fetal enterospheres (FEnS), is currently unknown and is most likely not epithelial (PMID: 24527386, PMID: 27117411 and PMID: 24821987). Instead, we have used RT-qPCR to assess the expression levels of *Axin2*: a commonly used readout of Wnt signalling activity (see, for example PMID: 26863187 or PMID: 21113151). Using FEnS/organoids cultures derived from small intestine we have found that, at a time point where the expression of several differentiation markers is only marginally reduced in *Ret51* mice, there is a strong reduction in *Axin2* transcript levels in *Ret51* cultures relative to controls. This data is now shown in Figure 6E and Figure EV4D. To investigate sufficiency, we have treated immature epithelium (pure populations of FEnS derived from E16.5 intestine) with GDNF: a Ret ligand. This treatment promotes branching and results in upregulation of *Axin2*. This data is now shown in Figures 6F-6H and, together with the *Ret51* loss-of-function data, shows that Ret signalling is both necessary and sufficient to promote organoid maturation and *Axin2* expression, further strengthening the links between Ret and Wnt signalling in mice.

Re: Src. In adult mouse intestine, Src is required for mouse intestinal tumourigenesis following upregulation of Wnt signalling (PMID: 24788409). The Jensen lab has a manuscript currently under review describing a Src requirement for Wnt-induced organoid maturation. Together with the experiments described above, these findings strengthen the links between Wnt and Src in the mouse intestine, lending further support to the model that Ret-induced Wnt/Src activation contributes to intestinal epithelial maturation. The Ret effects on *Axin2* and possible Src contributions to epithelial maturation in mice are now also discussed in more detail in the revised Discussion.

5. What is known about the mechanism of Ret signal transduction? What are the ligands in *Drosophila* and the mouse intestine? What are the downstream effectors? These questions should at least be discussed in more detail. Ideally, they would be addressed with data.

Regarding: ligands. The ligands that activate Ret may be different in flies and mammals. In mammals, Ret is activated by ligands of the glial cell line derived neurotrophic factor (Gdnf) family. During the development of the gastrointestinal tract, the epithelium of the small intestine would be exposed to the same mesenchymally derived Gdnf that ensures the survival, migration and differentiation of enteric neurons. Consistent with a contribution of Gdnf/Ret to intestinal epithelial maturation, we have exposed FEnS to GDNF and have confirmed that it can induce both branching and *Axin2* expression. This data is now shown in Figures 6F-6H.

In flies (which lack all Ret ligands belonging to the GDNF family), integrins bind to Ret to regulate nervous system development (PMID: 25764303). Interfering with integrin expression in the *Drosophila* intestine can have different effects on intestinal progenitor proliferation, survival and/or orientation depending on whether the integrins are removed from the progenitors or their niche - the visceral muscles (PMID: 23410794 and 23040479). Importantly, integrin downregulation in adult intestinal progenitors reduces their normal proliferation and can suppress their overproliferation in response to overactive Wingless signalling (PMID: 23410794 and our own observations, in Figure for Reviewers 3). These two phenotypes are strikingly similar to those resulting from Ret downregulation. Work currently under review in the Soba lab points to the existence of a second Ret ligand (a TGF-beta ligand) in the nervous system, but our expression analyses suggest that this ligand is not expressed in the adult midgut (data available upon request). Collectively, this data makes us favour the idea that integrins act as ligands for the Ret receptor in intestinal progenitors - perhaps by promoting contact between stem cells and enteroblasts: their undifferentiated, postmitotic progeny. In light of 1) the known links between integrins and Fak/Src signalling in both normal and cancer cells (PMID: 16919435) and 2) the effects that we have found for Src and Fak downstream of Ret activation, Ret could provide a new route for the integrin activation of the Src/Fak complex. These considerations are discussed in more detail in the revised discussion.

Re: downstream signalling. We have used RT-qPCR of dissected midguts to test whether gain- or loss-of-function of Ret in adult intestinal progenitors affects expression of a subset of candidate genes, selected because of their known effects on Wg signalling/Src activation. We have found no effects on the expression of these genes (Figure for Reviewers 1 and 4). We have, however, expanded our analysis of the Ret/Src interaction by exploring the contribution of Ret phosphorylation and the Ret/Src target Fak as described above (now shown in Figures 4A, F-H). We have also obtained new data concerning the JAK/STAT pathway. Further to the observation that lack of Ret does not render intestinal progenitors unable to respond to the JAK/STAT ligand upd1 (Figure 3A), we further find that the proliferative effects of Ret are, in fact, enhanced following downregulation of the JAK/STAT receptor *dome* or in mutants lacking two *upd* ligands (*updΔ2,3*) (Figure for Reviewers 5). This suggests that activation of the JAK/STAT pathway in intestinal progenitors by exogenous ligands provides an additional way to limit the proliferative effect of Ret. We find this finding interesting but, unlike the Fak break, it is harder to explain mechanistically in light of the known functions of the JAK/STAT pathway (previously shown to promote intestinal stem cell proliferation in response to damage). For this reason, we would prefer not to include in the revised manuscript unless the reviewer feels otherwise.

Referee #2

Perea and colleagues report the characterization of expression and function of the Ret receptor in intestinal stem cells of flies and mice. Using immunohistochemistry, genetic reporters, genetic perturbations and lineage tracing, the authors show that Ret is expressed in ISCs and EBs in the fly gut and that it is required for ISC proliferation in both homeostatic and stress conditions. Ret is also sufficient to induce ISC proliferation in a Wg-dependent manner. Increased Ret expression results in increased Wg expression, as well as increased phosphorylation of Src42A. Src42A, in turn, is required and sufficient to induce Wg expression.

The authors go on to assess the expression of Ret in the mouse intestine and find that it is highly expressed in the epithelium. Organoids derived from Ret deficient animals show branching defects, and Ret deficient tissue suggest that Ret is required for maturation of the intestinal epithelium.

The study is interesting, as it introduces a new receptor influencing the wg pathway in intestinal stem cells. It is well-designed and executed, and the presented results are clear and properly interpreted. Given the importance of understanding the control of epithelial homeostasis for cancer biology, it is clear that the study is of interest to a broad audience. There are a few concerns that the authors should address before publication can be recommended:

- The autoregulatory loop between Ret and Wg signaling is interesting, but also leads to difficulty in clearly establishing a mechanism for the role of Ret in ISCs. It may be useful to explore the activation of Wg expression and Src phosphorylation in a temporally resolved manner (for example in a time course after infection or DSS treatment). It can be anticipated that that would establish a temporal hierarchy of these events and may answer whether the interaction observed can be described as a linear pathway or whether there are parallel activation events of the two pathways that set up the autoregulatory pool.

We have attempted to conduct a time course experiment in *Drosophila* as suggested, both in response to DSS damage and bacterial (Ecc15) infection. In both cases, however, increased background staining - presumably resulting from epithelial damage - has made antibody detection very challenging (Figure for Reviewer 6). We now know that Ret upregulation of Wg is not transcriptional (see response to Reviewer 1, point 2, and that Src is activated by phosphorylation, so a transcriptional time course would be meaningless. See, however, our response to your next point re: the positive feedback loop and its break(s).

- The question of why this autoregulatory loop is set up is not addressed. It seems necessary to propose a mechanism by which the autoactivation is temporally restricted (otherwise triggering this pathway would immediately result in dramatic over-proliferation). Experiments to address this question would be interesting.

We have found that there is a non-cell autonomous component to the Ret-induced upregulation of Wg protein in adult intestinal progenitors. We have quantified *wg* transcript levels using RTqPCR following over-expression of Ret in adult-intestinal progenitors. In contrast to the *wg* transcript up-regulation detected following *wg* expression in these cells (positive control), Ret over-expression does not result in a significant increase of transcript levels. The observed lack of transcript induction suggests that the increased Wg protein observed following Ret expression may result from a non-transcriptional mechanism. To confirm this, we analysed *wg* transcription with cell specificity by making use of *wgKO-Gal4*: a *wg* transcription reporter in which Gal4 is expressed under the control of the endogenous *wg* promoter (Alexandre *et al* 2014 PMID: 24390349). We find that this reporter is not expressed in adult intestinal stem cells. Consistently, Ret misexpression from this *Gal4* driver fails to promote intestinal stem cell proliferation. These results, which are now shown in Figures 3D and E, are consistent with recent findings by Buchon *et al* (PMID:23643535) and Tian *et al* (PMID:23643535), which strongly suggest that *wg* expression occurs at intestinal compartment boundaries rather than adult intestinal progenitors. Together, both the published expression data and our functional experiments make us favour a model whereby Ret signalling in adult intestinal progenitors results in increased binding of Wg protein to these cells, possibly through upregulation/stabilisation of Wg receptor(s). As a result, Wg availability may limit Ret-driven proliferation.

We also hypothesize that, in normal homeostasis and regenerating guts, homeostasis-restoring mechanisms are deployed as part of the post-proliferative phase. These might be acting independently of Ret, but may ultimately contribute to prevent its proliferative actions. We have investigated genetically the contribution of two possible “proliferation breaks”: Fak and the Jak/Stat pathway.

Fak is a non-receptor tyrosine-protein kinases that can be phosphorylated by both Ret and Src kinases *ex vivo*, e.g. PMID: 21454698 and PMID: 8054685). We have found that Fak is phosphorylated in adult intestinal progenitors in response to Ret expression. Unlike Src, however, Fak does not promote Ret-induced proliferation; in fact, its co-expression with Ret prevents the Ret-induced proliferation increase. Src is still phosphorylated in this condition, suggesting that Fak blocks the mitogenic effect of Ret downstream of Src activation. These results, now shown in Figures 4F-H, lead us to hypothesize that, following Ret autophosphorylation and Ret/Src activation, these two kinases may phosphorylate and, consequently, activate Fak. Fak, in turn, would provide a break that limits the proliferative effects of the positive Ret/Wg feedback loop.

The new Wg and Fak data is now discussed in the context of limiting Ret-induced proliferation in the revised revised Discussion section.

We have also obtained new data concerning the JAK/STAT pathway. Further to the observation that lack of Ret does not render intestinal progenitors unable to respond to the JAK/STAT ligand upd1 (Figure 3A), we further find that the proliferative effects of Ret are, in fact, enhanced following downregulation of the JAK/STAT receptor *dome* or in mutants lacking two *upd* ligands (*updΔ2,3*) (Figure for Reviewers 5). This suggests that activation of the JAK/STAT pathway in intestinal progenitors by exogenous ligands provides an additional way to limit the proliferative effect of Ret. We find this finding interesting but, unlike the Fak break, it is harder to explain mechanistically in light of the known functions of the JAK/STAT pathway (previously shown to promote intestinal stem cell proliferation in response to damage). For this reason, we would prefer not to include in the revised manuscript unless the reviewer feels otherwise.

- The analysis in the mouse needs improvement. More careful characterization of Ret expression in a spatially resolved manner would help (which cells actually express Ret?)

We have extended the mouse expression data in both mouse small intestinal tissue and organoids. In particular, we have:

- Extended the *Ret* transcript analysis by providing RT-qPCR data obtained from *ex vivo* epithelial cultures (FEnS and organoids) to confirm *Ret* transcript expression in intestinal epithelium. This is now shown in Figure 5B.
- Characterised the relative abundance of *Ret* isoforms in both adult tissue and organoids. In both samples, ca. 75% of transcripts code for the *Ret9* isoform, further justifying the use of *Ret51* mice (lacking the *Ret9* isoform) for our functional studies. This is now shown in Figure EV3.
- Immunostained both adult small intestinal tissue and organoids with an anti-*Ret* antibody. Co-staining with an epithelial marker (E-cadherin) confirms that *Ret* is expressed in a subset of epithelial cells. This is now shown in Figures 5D and F (specificity of primary antibody shown in Figure EV2A).
- Co-stained both adult small intestinal tissue and organoids with *Ret* and cell-type specific markers (Paneth, goblet, enteroendocrine). This indicates that the subset of epithelial cells that express *Ret* are also positive for the enteroendocrine marker chromogranin-A. Costainings are now shown in Figures 5E, G and Figures EV2C and D. We have attempted similar stainings in FEnS and immature P1/P2 intestines, but have failed to detect a *Ret* antibody signal (Figure for Reviewers 2). We can, however, detect epithelial *Ret* transcript in immature epithelia, both by *in situ* hybridisation in neonatal intestine (Figure 5C) and RT-qPCR of epithelial FEnS cultures, where the levels may be lower than in organoids (Figure 5B). Hence, although we cannot identify a specific epithelial cell population expressing *Ret* at this stage (possibly because of lower expression), our transcriptional data points to epithelial *Ret* expression also in immature stages.

Together, this data confirms that *Ret* is expressed in mouse intestinal epithelium, and further indicates that *Ret* is expressed in a subset of secretory, chromogranin-A positive cells, present in both small intestinal tissue and organoids.

and experiments that address the mechanism by which *Ret* influences maturation (does it influence *Wg*? Does it affect Paneth cell differentiation?) are necessary.

We apologise that the effect on paneth cell differentiation, already provided in our initial manuscript, was not made sufficiently clear; we did not specifically state that Lysozyme was used as a Paneth cell marker). We have amended this.

Re: Wnt expression. We (the Jensen lab) have previously demonstrated that Wnt signalling drives intestinal maturation (PMID: 24139758). The observation that *Ret51* mutants display compromised maturation is aligned with these findings. To extend these observations, we have explored how *Ret* signalling affects Wnt signalling in gain- and loss-of-function experiments. We have not used Wnts as readouts because, in mice, there are multiple sources of intestinal Wnt ligands including the mesenchyme and Paneth cells of the intestinal epithelium (see, for example, PMID: 22922422 and PMID: 21113151). The relevant source of Wnt driving tissue maturation, observed as the appearance of organoids and disappearance of fetal enterospheres (FEnS), is currently unknown and is most likely not epithelial (PMID: 24527386, PMID: 27117411 and 24821987). Instead, we have used RT-qPCR to assess the expression levels of *Axin2*: a commonly used readout of Wnt signalling activity (see, for example PMID: 26863187 or PMID: 21113151). Using FEnS/organoids cultures derived from small intestine we have found that, at a time point where the expression of several differentiation markers is only marginally reduced in *Ret51* mice, there is a strong reduction in *Axin2* transcript levels in *Ret51* cultures relative to controls. This data is now shown in Figure 6E and Figure EV4D. To investigate sufficiency, we have treated immature epithelium (pure populations of FEnS derived from E16.5 intestine) with GDNF: a *Ret* ligand. This treatment promotes branching and results in upregulation of *Axin2*. This data is now shown in Figures 6F-6H and, together with the *Ret51* loss-of-function data, shows that *Ret* signalling is both necessary and sufficient to promote organoid maturation and *Axin2* expression, further strengthening the links between *Ret* and Wnt signalling in mice.

Referee #3

This manuscript describes the role of the Ret receptor in intestinal stem cells of *Drosophila*. The results are novel, solidly demonstrated and of significant interest to the community.

I have only one minor comment, which is that the positive feedback loop between Wg-Arm and Ret-Src seems to lack any proposed mechanism. How does Ret-Src activate Wg expression? And how does Wg-Arm signalling activate Ret-Src? Some additional evidence along these lines would strengthen the manuscript.

This is a difficult question because of the circular nature of the Wg/Src/Ret regulation. We have nonetheless attempted to address it as follows:

Ret signalling in intestinal progenitors increases Wg protein uptake, not *wg* transcription or *wg* transcript stability

We have quantified *wg* transcript levels using RT-qPCR following over-expression of Ret in adult intestinal progenitors. In contrast to the *wg* transcript up-regulation detected following *wg* expression in these cells (positive control), Ret over-expression does not result in a significant increase of transcript levels. The observed lack of transcript induction suggests that the increased Wg protein observed following Ret expression may result from a non-transcriptional mechanism. To confirm this, we analysed *wg* transcription with cell specificity by making use of *wgKO-Gal4*: a *wg* transcription reporter in which Gal4 is expressed under the control of the endogenous *wg* promoter (Alexandre *et al* 2014 PMID: 24390349). We find that this reporter is not expressed in adult intestinal stem cells. Consistently, Ret misexpression from this *Gal4* driver fails to promote intestinal stem cell proliferation. These results, which are now shown in Figures 3D and E, are consistent with recent findings by Buchon *et al* (PMID:23643535) and Tian *et al* (PMID:23643535), which strongly suggest that *wg* expression occurs at intestinal compartment boundaries rather than adult intestinal progenitors. Together, both the published expression data and our functional experiments make us favour a model whereby Ret signalling in adult intestinal progenitors results in increased binding of Wg protein to these cells, possibly through upregulation/stabilisation of Wg receptor(s). These considerations are now part of the revised discussion.

Links between Ret and Wnt signalling in mice

We have used RT-qPCR to assess the expression levels of Axin2: a commonly used readout of Wnt signalling activity (see, for example PMID: 26863187 or PMID: 21113151). Using FENS/organoids cultures derived from small intestine we have found that, at a time point where the expression of several differentiation markers is only marginally reduced in *Ret51* mice, there is a strong reduction in *Axin2* transcript levels in *Ret51* cultures relative to controls. This data is now shown in Figure 6E and Figure EV4D. Together with the GDNF-driven induction of *Axin2* expression and organoid branching mentioned above, these experiments shows that Ret signalling is both necessary and sufficient to promote organoid maturation and *Axin2* expression, further strengthening the links between Ret and Wnt signalling in mice.

Figures for Reviewers

Perea *et al.*

Figure for Reviewers 1. *Csk* transcript levels following expression of *wg*, *Ret* or *Ret-RNAi* from adult intestinal progenitors

Figure for Reviewers 2 Lack of *Ret* expression in the epithelium of P2 small intestinal tissue

Figure for Reviewers 3. Downregulation of *mys* and *mew* integrins in adult progenitors abrogates *Ret*-induced proliferation and *Wg* upregulation

Figure for Reviewers 4. *swim* transcript levels following expression of *Ret* or *Ret-RNAi* from adult intestinal progenitors

Figure for Reviewers 5. *upd* mutation or *dome-RNAi* co-expression increases the pro-proliferative effect of *Ret* expression in adult intestinal progenitors

Figure for Reviewers 6. Reduced quality of *Ret* antibody staining following DSS-induced epithelial damage

Thank you for submitting your revised manuscript for consideration by The EMBO Journal, and your patience with our response. Your revised study was sent back to all three referees for re-evaluation. However, referee #2 was delayed and has not sent us his/her report. In the interest of time, we have now decided to continue with our decision and have thus editorially assessed if his/her criticism were adequately addressed. Please find the comments of the two other referees enclosed below.

As you will see, the first referee remains overall more critical on the study, however we decided - in light of the strong support of the other referees - to give you the opportunity to revise your manuscript to address the referee's points.

In more detail, referee #3 finds that his/her concerns have been sufficiently addressed and is in broadly favour of publication. Referee #1 agrees on novelty and robustness of your study, but in addition states, that your claims on direct causality between Ret/Src/Wg/Fak signaling and Ret-activation of ISC proliferation are not sufficiently well supported by the current data (ref #1, pt. 1). This referee also points out that there is a need for you explore MAPK/ERK signaling as an effector of Ret in ISCs (ref #1, pt. 2), as well as to revise your mutant analysis of mouse intestine (ref #1, pt. 3). Please note, however, that considering the positive comments of referee #3 and referee #1 in his/her earlier report, we have decided that pending a satisfactory revision, we would go ahead with acceptance of this manuscript as soon as possible. Thus, I ask you to revise your manuscript regarding the points raised by referee #1 and evaluate, whether you would be able to add complementary data, or, alternatively, relativise your statements and introduce caveats where appropriate. I also encourage you to add references in the discussion to the related recent findings by Bellono et al, 2017.

REFEREE REPORTS

Referee

#1:

As in the earlier version, this paper presents a substantial body of data relevant to the function of the receptor tyrosine kinase, Ret, in the Drosophila and Mouse intestinal epithelia. The data presented are good quality and the specific conclusions are mostly reasonable. Based on their data, the authors first propose that Ret function is autonomously required in Drosophila intestinal stem cells for their normal proliferation. This conclusion is solidly supported, and it's a valuable message. However the rest of the paper delivers results with less clear-cut meaning, and I think it could still benefit from some modest revisions, including new data, as described below.

1. From their work in Drosophila, the authors conclude that Ret works via a wg/Fak/Src signaling system that's rather novel. This is supported by expression and epistasis tests, but no data relevant to the molecular nature of the cross regulation between these gene functions is provided, and it seems the roles of Wg, Src, and Fak could be rather indirect. In discussing these experiments the authors make the common mistake of concluding, from epistasis tests and expression alone, that the up-regulation of Wg, Src and Fak activity by Ret is required for Ret to activate ISC proliferation. While many authors make such conclusions based on similar data, it's an over-interpretation that is not well justified. In fact the situation might be that though Src or Sgg (for instance) are required for Ret-induced proliferation, the the Ret-induced increases in Wg expression and Src phosphorylation are actually functionally irrelevant. Epistasis experiments like those shown in Figs 3-4 test only the requirement for a gene or gene product for a process, and can't actually test the relevance of a change in gene activity. Unfortunately tests of this are not simple or obvious. Nevertheless, this is not a subtle point and I feel the authors should change the declarative section titles on pages 4 and 5, which have over-stated the definitiveness of the epistasis tests in Figs 3 and 4.

2. Given that Ret is widely believed to signal via the Ras/Raf/Erk pathway, and that these factors are known to be essential for Drosophila ISC proliferation, the omission of tests of the MAPK pathway as an effector of Ret is quite disappointing. If they did such tests, the authors could well find that Ret also induces ERK activity, and that Ras, Raf, Mek and Erk are required (like Wg and Src) for Ret-

induced ISC proliferation. I think the authors should test this obvious possibility. This would help to fill out the picture of how Ret controls ISC proliferation, and it might also help inform the responses seen with Src, Fak, and Wg.

3. As in the previous version, the analysis of Ret expression and function in the mouse intestine is quite preliminary, and no major conclusions can be derived from it. The paper does show that Ret is expressed there, and may have a function in maturation (a vaguely defined term), but little more can be said. The mutant analysis is confounded by the presence of several Ret paralogs (only one was removed here), and the suggestion that Ret regulates Wnt signaling is based on preliminary, indirect assays that are not at all analogous to the ligand expression and epistasis tests performed in flies. The mouse experiments (Fig 5-6) don't stand alone with clear message, and also don't much in terms of supporting the more extensive analysis in *Drosophila*. I'm not sure precisely what to suggest here, but a deeper functional analysis of Ret would certainly be welcome.

Referee #3:

The mechanism of the feedback loop between Wg and Ret still seems a little unclear, but overall I am satisfied with the manuscript.

2nd Revision - authors' response

26 July 2017

Perea et al Ret in intestinal epithelia

Response to Referee #1 *As in the earlier version, this paper presents a substantial body of data relevant to the function of the receptor tyrosine kinase, Ret, in the Drosophila and Mouse intestinal epithelia. The data presented are good quality and the specific conclusions are mostly reasonable. Based on their data, the authors first propose that Ret function is autonomously required in Drosophila intestinal stem cells for their normal proliferation. This conclusion is solidly supported, and it's a valuable message. However the rest of the paper delivers results with less clear-cut meaning, and I think it could still benefit from some modest revisions, including new data, as described below.*

We are glad that the referee regards our data as solid, substantial and valuable. As explained below in more detail, we have inserted additional clarifications including two recent publications that help interpret the “less clear-cut” data that the referee alludes to.

1. From their work in Drosophila, the authors conclude that Ret works via a wg/Fak/Src signaling system that's rather novel. This is supported by expression and epistasis tests, but no data relevant to the molecular nature of the cross regulation between these gene functions is provided, and it seems the roles of Wg, Src, and Fak could be rather indirect. In discussing these experiments the authors make the common mistake of concluding, from epistasis tests and expression alone, that the up-regulation of Wg, Src and Fak activity by Ret is required for Ret to activate ISC proliferation. While many authors make such conclusions based on similar data, it's an over-interpretation that is not well justified. In fact the situation might be that though Src or Sgg (for instance) are required for Ret-induced proliferation, the Ret-induced increases in Wg expression and Src phosphorylation are actually functionally irrelevant. Epistasis experiments like those shown in Figs 3-4 test only the requirement for a gene or gene product for a process, and can't actually test the relevance of a change in gene activity. Unfortunately tests of this are not simple or obvious. Nevertheless, this is not a subtle point and I feel the authors should change the declarative section titles on pages 4 and 5, which have over-stated the definitiveness of the epistasis tests in Figs 3 and 4.

As explained in the revised version of the manuscript, previous work has demonstrated direct binding/phosphorylation of Ret/Src and Src/Fak. This data is mentioned and referenced in the revised manuscript (in paragraphs 1 and 2 of Src42A/Fak Results section). Whilst we agree that epistasis and expression experiments alone do not prove that up-regulation of Ret is required to activate proliferation, we do show that 1) Ret is normally expressed in adult progenitors (antibody and Gal4 data) and 2) its mutation impairs proliferation, both in homeostatic and regenerative conditions and 3) its over-expression in adult progenitors

increases proliferation. Based on these findings, it seems reasonable to conclude that Ret is required in adult progenitors to control proliferation. Both our work and that of others (e.g. Cordero et al, 2012; Lin et al, 2008; Tian et al, 2016; Cordero et al, 2014; Kohlmaier et al, 2015) have carried out similar experiments to infer similar functions for Wg and Src. We do, however, agree that, from these experiments, one cannot establish the sequence of signalling events in a more physiological context (namely, the molecular trigger up- or downregulated following, for example, epithelial damage and the ensuing cellular events). Our manuscript (including the titles the referee alludes to) did not make this claim but, in this revised manuscript, we have 1) replaced “controls” with “promotes/affects” in two Results titles to better reflect the phenotypes described in Figs 3 and 4, and 2) amended a relevant sentence in the first paragraph of the Discussion to “Our gain- and loss-of-function experiments point to the existence of positive feedback between Ret and Wg signalling”

2. Given that Ret is widely believed to signal via the Ras/Raf/Erk pathway, and that these factors are known to be essential for Drosophila ISC proliferation, the omission of tests of the MAPK pathway as an effector of Ret is quite disappointing. If they did such tests, the authors could well find that Ret also induces ERK activity, and that Ras, Raf, Mek and Erk are required (like Wg and Src) for Ret-induced ISC proliferation. I think the authors should test this obvious possibility. This would help to fill out the picture of how Ret controls ISC proliferation, and it might also help inform the responses seen with Src, Fak, and Wg.

It will, indeed, be of interest to explore signalling downstream of Ret receptor activation further, and MAPK is an obvious candidate. We believe, however, that this is beyond the scope of this manuscript, but have inserted a sentence in the second Discussion section to acknowledge this relevant consideration.

3. As in the previous version, the analysis of Ret expression and function in the mouse intestine is quite preliminary, and no major conclusions can be derived from it. The paper does show that Ret is expressed there, and may have a function in maturation (a vaguely defined term), but little more can be said. The mutant analysis is confounded by the presence of several Ret paralogues (only one was removed here), and the suggestion that Ret regulates Wnt signaling is based on preliminary, indirect assays that are not at all analogous to the ligand expression and epistasis tests performed in flies. The mouse experiments (Fig 5-6) don't stand alone with clear message, and also don't much in terms of supporting the more extensive analysis in Drosophila. I'm not sure precisely what to suggest here, but a deeper functional analysis of Ret would certainly be welcome.

Whilst we agree that future studies in which Ret is knocked out in vivo from intestinal epithelial cells will be very informative (and have stated this in the revised discussion), we disagree regarding “confounds”. The function in organoid maturation (a term referring to the transition from foetal to mature epithelium, see for example Fordham et al, 2013) is well documented in Figs. 6 and EV4. There are no Ret “paralogues”. There is only one Ret gene, which is spliced into two different isoforms (Fig. EV3). We have used Ret51 mice that lack the Ret9 isoform: the most abundant form in intestinal epithelium (Fig. 6A). Because mice lacking both isoforms are lethal, the use of this mutant allows us to recover intestinal tissue that lacks >70% of Ret function in intestinal epithelium. We have been able to observe a phenotype in this mice, so we anticipate that complete absence of Ret in intestinal epithelium will be as, if not more, severe.

Re: Ret/Wnt in mice. Although further work will be required to establish where and how the interaction between Ret and Wnt signalling occurs in mice (an issue that is not fully resolved even in the case of Wnt ligands/signalling alone), we do not feel our assays are “indirect”. Indeed, we have provided not only gain-of-function but also loss-of-function experiments that lend support to the idea that Ret signalling affects Wnt signalling, and provide evidence in both cases that Axin2 (a routinely used readout of Wnt signalling in small intestine/organoids, Clevers et al, 2014; Fordham et al, 2013; Jho et al, 2002; van Es et al, 2005) is preferentially affected relative to other epithelial cell markers.

In addition to this earlier role in epithelial maturation, we find that Ret expression is maintained in a subset of chromogranin A cells. This is exciting because a very recent paper has suggested that cells expressing enteroendocrine markers including chromogranin A can

have stem cell activity in the adult epithelium (Yan et al, 2017). This is exciting because it suggests that, whilst the cellular classification of Ret-positive cells based on known markers may differ between flies (ISCs) and mice (enteroendocrine), Ret-enabled stem cell functionality may contribute to regeneration in both epithelia. For this reason, we believe that the mouse data synergises with - rather than weakens - the fly data. We have discussed this recent paper in the first section of the revised discussion, together with the other recent and relevant finding that these chromogranin A-positive cells are chemosensory (Bellono et al, 2017).

Corresponding Author Name: Irene Miguel-Aliaga

Journal Submitted to: EMBO J

Manuscript Number: EMBOJ-2016-96247R